# Novel axonemal protein ZMYND12 interacts with TTC29 and DNAH1, and is required for male fertility and flagellum function

Denis Dacheux[1,2†], Guillaume Martinez[3†], Christine E Broster Reix[1], Julie Beurois[4], Patrick Lores[5], Magamba Tounkara[1], Jean-William Dupuy[6], Derrick Roy Robinson[1], Corinne Loeuillet[4], Emeline Lambert[4], Zeina Wehbe[4], Jessica Escoffier[4], Amir Amiri-Yekta[7], Abbas Daneshipour[7], Seyedeh-Hanieh Hosseini[8], Raoudha Zouari[9], Selima Fourati Ben Mustapha[9], Lazhar Halouani[9], Xiaohui Jiang[10,11,12], Ying Shen[11,12], Chunyu Liu[13], Nicolas Thierry-Mieg[14], Amandine Septier[14], Marie Bidart[4,15], Véronique Satre[3,4], Caroline Cazin[3,4,16], Zine Eddine Kherraf[4,16], Christophe Arnoult[4], Pierre F Ray[4,16], Aminata Toure[17‡], Mélanie Bonhivers[1‡], Charles Coutton[4*‡]

[1]University of Bordeaux, CNRS, Bordeaux, France; [2]Bordeaux INP, Microbiologie Fondamentale et Pathogénicité, Bordeaux, France; [3]CHU Grenoble-Alpes, UM de Génétique Chromosomique, Grenoble, France; [4]Institute for Advanced Biosciences, INSERM U1209, CNRS UMR 5309, Université Grenoble Alpes, Team Genetics Epigenetics and Therapies of Infertility, Grenoble, France; [5]Institut Cochin, INSERM U1016, CNRS UMR 8104, Université Paris Cite, Paris, France; [6]Université Bordeaux, Plateforme Protéome, Bordeaux, France; [7]Department of Genetics, Reproductive Biomedicine Research Center, Royan Institute for Reproductive Biomedicine, ACECR, Tehran, Islamic Republic of Iran; [8]Department of Andrology, Reproductive Biomedicine Research Center, Royan Institute for Reproductive Biomedicine, ACECR, Tehran, Islamic Republic of Iran; [9]Polyclinique les Jasmins, Centre d'Aide Médicale à la Procréation, Centre Urbain Nord, Tunis, Tunisia; [10]Human Sperm Bank, West China Second University Hospital of Sichuan University, Sichuan, China; [11]NHC Key Laboratory of Chronobiology, Sichuan University, Sichuan, China; [12]Key Laboratory of Birth Defects and Related Diseases of Women and Children (Sichuan University), Ministry of Education, Sichuan, China; [13]Obstetrics and Gynecology Hospital, Fudan University, Fudan, China; [14]Université Grenoble Alpes, CNRS, Grenoble, France; [15]CHU Grenoble Alpes, Laboratoire de Génétique Moléculaire: Maladies Héréditaires et Oncologie, Grenoble, France; [16]CHU de Grenoble, UM GI-DPI, Grenoble, France; [17]Institute for Advanced Biosciences, INSERM U 1209, CNRS UMR 5309, Université Grenoble Alpes, Team Physiology and Pathophysiology of Sperm cells, Grenoble, France

**\*For correspondence:**
ccoutton@chu-grenoble.fr

†These authors contributed equally to this work
‡These authors also contributed equally to this work

**Competing interest:** The authors declare that no competing interests exist.

**Abstract** Male infertility is common and complex, presenting a wide range of heterogeneous phenotypes. Although about 50% of cases are estimated to have a genetic component, the underlying cause often remains undetermined. Here, from whole-exome sequencing on samples from 168 infertile men with asthenoteratozoospermia due to severe sperm flagellum, we identified homozygous *ZMYND12* variants in four unrelated patients. In sperm cells from these individuals,

immunofluorescence revealed altered localization of DNAH1, DNALI1, WDR66, and TTC29. Axonemal localization of ZMYND12 ortholog TbTAX-1 was confirmed using the *Trypanosoma brucei* model. RNAi knock-down of TbTAX-1 dramatically affected flagellar motility, with a phenotype similar to the sperm from men bearing homozygous *ZMYND12* variants. Co-immunoprecipitation and ultrastructure expansion microscopy in *T. brucei* revealed TbTAX-1 to form a complex with TTC29. Comparative proteomics with samples from *Trypanosoma* and *Ttc29* KO mice identified a third member of this complex: DNAH1. The data presented revealed that ZMYND12 is part of the same axonemal complex as TTC29 and DNAH1, which is critical for flagellum function and assembly in humans, and *Trypanosoma*. ZMYND12 is thus a new asthenoteratozoospermia-associated gene, bi-allelic variants of which cause severe flagellum malformations and primary male infertility.

## eLife assessment

This **important** study reports the physiological role of ZMYND21 in the regulation of sperm flagellar development and male fertility. The data supporting the conclusion are **solid**, although the inclusion of more patients and ultrastructural studies would have further strengthened the study. This work will be of interest to clinicians and researchers who work on either sperm biology or ciliopathy due to cilial defects.

## Introduction

Male infertility is one of the most challenging health issues that will face us in the near future. Recent decades have been marked by a significant and constant deterioration of quantitative and qualitative sperm parameters in both humans and animals, demonstrating the detrimental impact of recent environmental changes (*Rolland et al., 2013*). However, environmental factors are not the sole cause of male infertility, which is often multifactorial. Among the various possible causes, it is estimated that a genetic component, alone or in association, can be found in about half of cases (*Oud et al., 2022*).

Spermatozoa are the most highly differentiated and specialized human cells, as reflected by the 2300 genes necessary for normal spermatogenesis (*Lu et al., 2019*). The complex process of spermatogenesis is divided into three phases: proliferation and differentiation of spermatogonia, meiosis, and the final critical step, spermiogenesis. During spermiogenesis, haploid round spermatids undergo an extraordinary series of transformations to acquire the specific morphological features necessary for them to competently fertilize an egg. The stages in spermiogenesis include nuclear compaction, acrosome formation, and flagellum biogenesis (*O'Donnell, 2014*). Any defects in genes driving these differentiation steps can lead to severe functional or morphological sperm defects, and thus male infertility.

Interestingly, the prevalence of genetic defects is higher in conditions with severe sperm phenotypes (*Coutton et al., 2015*). Multiple Morphological Abnormalities of the sperm Flagellum (MMAF) is a condition associated with extreme morphological sperm defects characterized by a mosaic of phenotypes. Thus, MMAF sperm cells may lack a flagellum or bear a short, irregular, or coiled flagellum. Because of these characteristics, MMAF causes total asthenozoospermia with almost no progressive sperm (*Touré et al., 2021*). At the ultrastructural level, MMAF is associated with severe disorganization of both axonemal and peri-axonemal structures (*Nsota Mbango et al., 2019*). Approximately 40 genes have been strongly or definitively associated with the MMAF phenotype so far (*Wang et al., 2020*). However, this high genetic heterogeneity means the prevalence of individual variants of each gene remains low, and the underlying genetic component remains unknown for about half of affected individuals. It is therefore necessary to pursue the identification of new candidate MMAF genes to increase the diagnostic yield and improve our understanding of the pathophysiological mechanisms inducing this sperm phenotype. In addition, the identification of MMAF genes provides invaluable clues to the molecular mechanisms underlying sperm flagellum biogenesis. Therefore, efforts are continuously being deployed to decipher the precise organization, localization, and interactions of each protein encoded by these genes within the flagellum.

Here, following analysis of an initial cohort of 167 MMAF individuals by whole-exome sequencing (WES), we identified truncating homozygous variants in *ZMYND12* in three unrelated individuals. A fourth individual with a similar mutation was subsequently identified in a Chinese cohort. In this paper,

we present a thorough investigation of the role of this new gene by studying its ortholog, TbTAX-1, in *Trypanosoma*, an evolutionarily distant model sharing a highly conserved flagellar structure. We first showed the importance of ZMYND12 for the axonemal structure and function of the flagellum. Subsequently, we shed light on ZMYND12 function in the flagellum using the TbTAX-1 protist model and *Tttc29* KO mice. Results formally demonstrated that ZMYND12 interacts directly with DNAH1 and TTC29 (two known MMAF-related proteins) within the same axonemal complex.

## Results

### WES identified homozygous truncating variants in *ZMYND12* in MMAF individuals

In our cohort of 167 MMAF individuals, we previously identified 83 individuals (49.7%) with harmful gene variants in known MMAF-related genes. Reanalysis of the remaining negative exomes identified three unrelated MMAF individuals (ZMYND12$_{1-3}$) (*Figure 1A*, *Table 1*) carrying homozygous variants in *ZMYND12* (Zinc Finger MYND-Type Containing 12, NM_032257.5). This gene was not previously associated with any pathology. Two individuals (ZMYND12$_1$ and ZMYND12$_3$) originated from Tunisia, and one individual (ZMYND12$_2$) was of Iranian origin.

*ZMYND12* is located on chromosome 1 and contains eight exons encoding the Zinc Finger MYND domain-containing protein 12, a predicted 365-amino acid protein (Q9H0C1). According to data from GTEx, *ZMYND12* is strongly expressed in the testis and is associated with cilia and flagella (*Yamamoto et al., 2021*; *Yamamoto et al., 2006*). Moreover, quantitative single-cell RNA sequencing datasets from human adult testis (ReproGenomics Viewer) (*Darde et al., 2019*) indicate abundant expression in germ cells from zygotene spermatocyte to late spermatid stages. This expression pattern suggests a role in sperm cell differentiation and/or function. ZMYND12 protein was also detected at significant levels in the human sperm proteome (*Wang et al., 2013*), but at very low levels in human airway cilia (*Blackburn et al., 2017*). We performed quantitative real-time RT-PCR (RT-qPCR) experiments on human tissue panels (*Figure 1—figure supplement 1*), and results confirmed that expression of *ZMYND12* mRNA is predominant in testis as compared with the other tissues tested.

A stop-gain variant c.433C>T; p.Arg145Ter (NM_032257.5, GRCh38 g.chr1:42440017G>A, rs766588568), located in *ZMYND12* exon 4 (*Figure 1B*), was identified in individuals ZMYND12$_1$ and ZMYND12$_3$. Subsequently, a Chinese MMAF subject harboring the same homozygous stop-gain variant in *ZMYND12* (ZMYND12$_4$) was identified in the cohort collected by the Human Sperm Bank of West China Second University Hospital of Sichuan University hospital (*Figure 1*, *Table 1*). This c.433C>T variant is present in the Genome Aggregation Database (gnomAD v2.1.1, http://gnomad. broadinstitute.org/variant/1-42905688-G-A) database with a minor allele frequency of 3.391e−05. This nonsense variant, identified by exome sequencing, was confirmed by Sanger sequencing in all four individuals (*Figure 1C*).

Using the ExomeDepth software package (*Plagnol et al., 2012*), individual ZMYND12$_2$, was identified based on a homozygous deletion removing *ZMYND12* exons 6–8 (*Figure 1—figure supplement 2*). We confirmed the presence of the homozygous deletion in the affected individual by custom MLPA (*Figure 1D*). The deletion identified was not listed in public databases, including DECIPHER (https://www.dechipergenomics.org/) and the Database of Genomic Variants (http://dgv.tcag.ca/dgv/app/home).

For the individuals carrying *ZMYND12* mutations, analysis of control databases revealed no low-frequency variants in other genes reported to be associated with cilia, flagella, or male fertility. We therefore focused our study on *ZMYND12*, which appeared as the best MMAF candidate gene. All of the *ZMYND12* variants identified here have been deposited in ClinVar under reference SUB12540308.

### ZMYND12 is located in the sperm flagellum, its variants cause severe axonemal disorganization

To further investigate the pathogenicity of the *ZMYND12* variants identified, we examined the distribution of the ZMYND12 protein in control and mutated sperm cells by immunofluorescence (IF). Due to sample availability, these analyses were only carried out for individual ZMYND12$_3$ – carrier of the recurrent stop-gain variant c.433C>T. In sperm from control individuals, ZMYND12 antisera decorated the full length of the sperm flagellum (*Figure 1E*). Conversely, and in line with a truncation or

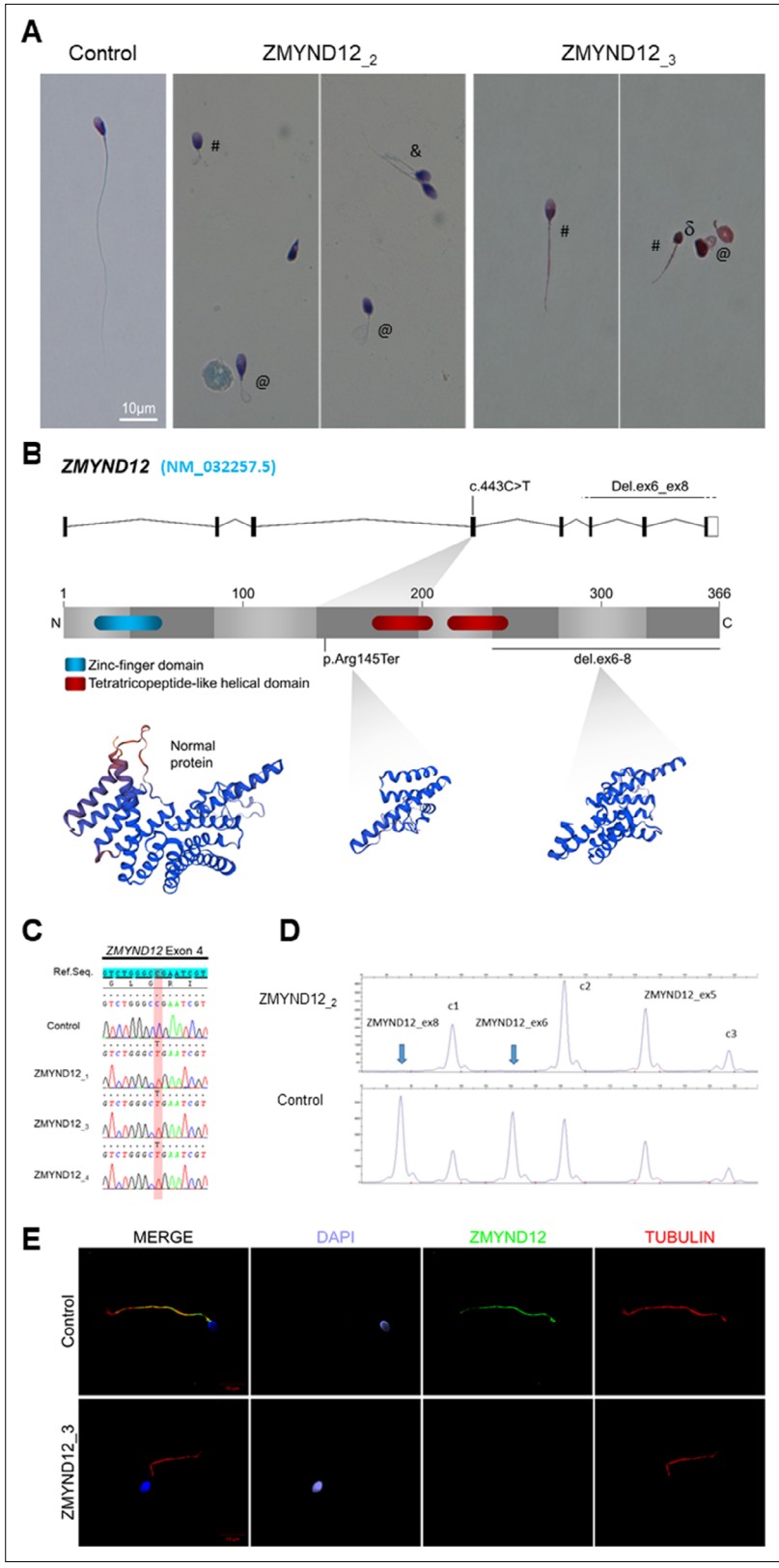

**Figure 1.** Morphology of normal and *ZMYND12* mutant spermatozoa and variants identified in *ZMYND12* individuals. (**A**) Light microscopy analysis of spermatozoa from fertile control individuals and from individuals ZMYND12_2 and ZMYND12_3. Most spermatozoa from ZMYND12 individuals had short (#), coiled (@), or irregular caliber (&) flagella. Head malformations were also observed (δ). (**B**) Location of the variants identified within the

*Figure 1 continued on next page*

*Figure 1 continued*

*ZMYND12* gene (NM_032257.5) and protein according to UniprotKB (Q9H0C1). The blue box represents a Zinc Finger domain and the orange boxes represent TPR domains (tetratricopeptide repeat). (**C**) Sanger sequencing electropherograms indicating the homozygous state of the *c.433C>T* variant located in exon 4 in the three individuals ZMYND12_1,3 and 4. The substituted nucleotide is highlighted in red. Variants are annotated in line with HGVS recommendations. (**D**) Confirmation of *ZMYND12* exon deletion by MLPA. MLPA profiles obtained for control and ZMYND12_2 individuals. Each set contains three control probes for normalization purposes (c1–c3) and three *ZMYND12*-specific probes. Blue arrows indicate homozygous deletion of exons 6 and 8 in the *ZMYND12* gene. *ZMYND12* exon 5 is not included in the deletion. (**E**) Sperm cells from a fertile control individual and from individual ZMYND12_3 – carrier of the *c.433C>T* variant – were stained with anti-ZMYND12 (green) and anti-acetylated tubulin (red) antibodies. Sperm nuclear DNA was counterstained with DAPI (4',6-diamidino-2-phenylindole) (blue). In control sperm, ZMYND12 immunostaining was present along the whole length of the flagellum; no staining was observed in sperm cells from individual ZMYND12_3. Scale bars: 10 μm.

The online version of this article includes the following figure supplement(s) for figure 1:

**Figure supplement 1.** Relative mRNA expression of human *ZMYND12* transcripts.

**Figure supplement 2.** Depth of coverage at the *ZMYND12* locus for the ZMYND12_2 individual, carrier of a homozygous deletion of exons 6–8, and four other unrelated Multiple Morphological Abnormalities of the sperm Flagellum (MMAF) individuals.

**Figure supplement 3.** ZMYND12 immunostaining in human spermatozoa from Multiple Morphological Abnormalities of the sperm Flagellum (MMAF) individuals carriers of pathogenic variants in *ARMC2* and *WDR66* genes, or with unknwon genetic cause.

degradation of the mutated protein, ZMYND12 was totally absent from all sperm cells from individual ZMYND12_3 (*Figure 1E*). Importantly, ZMYND12 was detected in sperm cells from MMAF individuals bearing mutations in other genes (*ARMC2* and *WDR66*) or with unknown genetic causes (*Figure 1—figure supplement 3*). Thus, the absence of ZMYND12 is not a common feature of the MMAF phenotype, and is specifically associated with *ZMYND12* variants.

To explore the ultrastructural defects induced by *ZMYND12* variants in human sperm cells, we next studied the presence of proteins present in various axonemal and peri-axonemal substructures by IF. The following proteins were investigated: AKAP4 (fibrous sheath, FS), CFAP70 (axoneme-binding protein with unknown precise location), DNAI1 (outer arm intermediate chain), DNALI1 (inner arm light intermediate chain), DNAH1 (inner arm heavy chain), DNAH8 (outer arm heavy chain), DNAH17 (outer arm heavy chain), GAS8 (Nexin-Dynein Regulatory Complex, N-DRC), RSPH1 (radial spoke, RS), SPAG6 (central pair complex, CPC), WDR66/CFAP251 (calmodulin- and spoke-associated complex, CSC), and TTC29 (axoneme-binding protein with unknown precise location). In sperm cells from ZMYND12_3, we first observed a total lack of SPAG6 staining (*Figure 2—figure supplement 1A*), suggesting severe CPC defects. This type of defect is a common feature of MMAF spermatozoa (*Coutton et al., 2015*). In addition, no immunostaining for DNAH1, DNALI1, or WDR66 was detected in sperm from individual ZMYND12_3 (*Figure 2A, B* and *Figure 2—figure supplement 1B*). These results suggested a strong disorganization of the IDAs (Inner dynein arms) and the CSC. ZMYND12 could therefore be a key axonemal component responsible for stability of these structures.

We subsequently examined the presence of TTC29 and CFAP70, two proteins with as yet incompletely characterized function and localization. In control sperm, TTC29 and CFAP70 immunostaining decorated the full length of the flagellum, but neither protein was detected in cells from individual ZMYND12_3 (*Figure 2C, D*). Intriguingly, ZMYND12 was undetectable in the sperm flagellum from an MMAF individual carrying a previously reported *TTC29* splicing variant c.1761+1G>A (*Lorès et al., 2019*; *Figure 2—figure supplement 2*). Unfortunately, no sample from an individual carrying a bi-allelic *CFAP70* variant could be analyzed. Nevertheless, the available data suggest that TTC29, CFAP70, and ZMYND12 may be physically or functionally associated within the same axonemal complex.

We also observed heterogeneous RSPH1 staining with all flagellar morphologies (*Figure 2—figure supplement 3*), indicating that RSs are inconsistently present in sperm cells from individual ZMYND12_3. In contrast, the immunostaining patterns for AKAP4, DNAH8, DNAH17, DNAI1, and GAS8 were comparable to those observed in control sperm cells. This result indicates that the FS, the outer dynein arms (ODAs), and the N-DRC were not directly affected by *ZMYND12* variants (*Figure 2—figure supplement 4*). Transmission electron microscopy (TEM), which could provide

**Table 1.** Detailed semen parameters in the four Multiple Morphological Abnormalities of the sperm Flagellum (MMAF) individuals harboring a *ZMYND12* variant.

*ZMYND12* mutated individuals

| Individuals | Homozygous ZMYND12 variants | Semen parameters | | | | | | | | | | | | | | | | |
|---|---|---|---|---|---|---|---|---|---|---|---|---|---|---|---|---|---|---|
| | | Sperm volume (ml) | Sperm concentration ($10^6$/ml) | Total motility 1 hr | Vitality | Normal spermatozoa | Absent flagella | Short flagella | Coiled flagella | Bent flagella | Flagella of irregular caliber | Tapered head | Thin head | Micro-cephalic | Macro-cephalic | Multiple heads | Abnormal base | Abnormal acrosomal region |
| ZMYND12$_1$ | c.433C>T | 2 | 6.5 | 4 | 70 | 0 | 35 | 48 | 10 | 0 | 34 | 22 | 2 | 2 | 4 | 6 | 20 | 80 |
| ZMYND12$_2$ | Del ex6-8 | 2.5 | 4.25 | 2 | NA | 0 | NA | 72 | NA | NA | NA | NA | NA | NA | NA | NA | NA | NA |
| ZMYND12$_3$ | c.433C>T | 3 | 34 | 2 | 50 | 0 | 20 | 20 | 28 | 0 | 46 | 22 | 6 | 8 | 2 | 2 | 24 | 88 |
| ZMYND12$_4$ | c.433C>T | 2.6 | 0.1 | 0 | NA | NA | NA | 70 | NA | NA | NA | NA | NA | NA | NA | NA | NA | NA |
| Reference limits* | | 1.5 (1.4-1.7) | 15 (12-16) | 40 (38-42) | 58 (55-63) | 23 (20-26) | 5 (4-6) | 1 (0-2) | 17 (15-19) | 13 (11-15) | 2 (1-3) | 3 (2-4) | 14 (12-16) | 7 (5-9) | 1 (0-2) | 2 (1-3) | 42 (39-45) | 60 (57-63) |

Values are percentages unless specified otherwise. NA: not available.

*Reference limits (5th centiles and their 95% confidence intervals) according to World Health Organization (WHO) standards (**Cooper et al., 2010**) and the distribution range of morphologically normal spermatozoa observed in 926 fertile individuals (**Auger et al., 2016**).

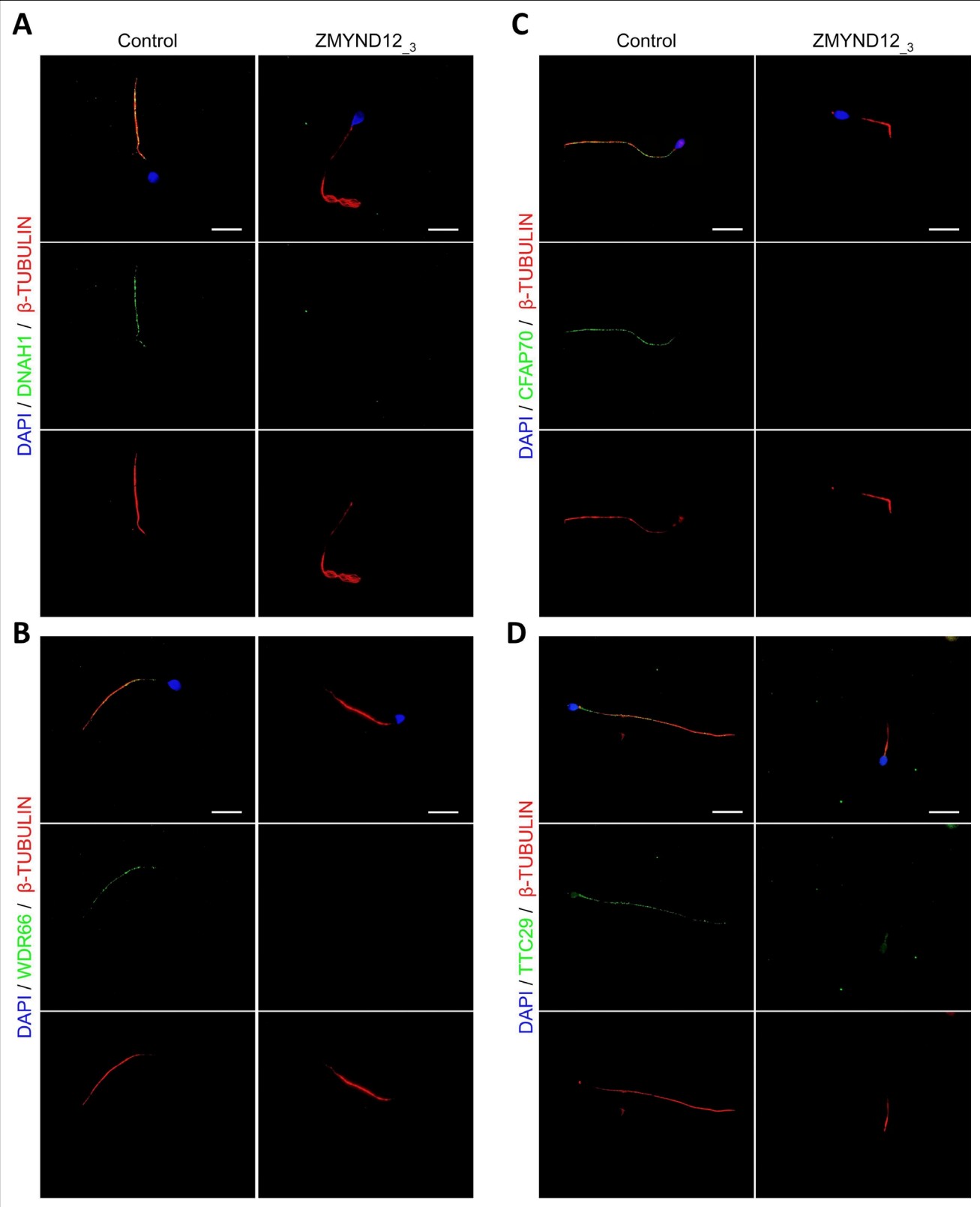

**Figure 2.** Altered immunostaining for DNAH1, WDR66, CFAP70, and TTC29 in the presence of *ZMYND12* variants. Immunofluorescence experiments were performed using sperm cells from control individuals and from individual ZMYND12_3 carrying the nonsense variant *c.433C>T*. Although tubulin staining remained detectable, sperm cells from individual ZMYND12_3 showed no immunostaining for (**A**) DNAH1, (**B**) WDR66, (**C**) CFAP70, and (**D**) TTC29. Scale bars: 10 μm.

*Figure 2 continued on next page*

*Figure 2 continued*
The online version of this article includes the following figure supplement(s) for figure 2:
**Figure supplement 1.** Altered immunostaining for SPAG6 and DNALI1 in the presence of *ZMYND12* variants.
**Figure supplement 2.** ZMYND12 is absent from sperm cells from *TTC29* individuals.
**Figure supplement 3.** Radial spokes are inconstantly affected in *ZMYND12* individuals.
**Figure supplement 4.** Immunostaining for AKAP4, DNAH2, DNAH8, DNAH17, DNAI1, and GAS8 is not affected by *ZMYND12* mutation.

direct evidence of ultrastructural defects, could not be performed due to the very low number of sperm cells available from *ZMYND12* individuals.

## TbTAX-1, the *Trypanosoma brucei* ortholog of ZMYND12, is an axoneme-associated protein involved in flagellar motility

To validate our candidate gene and better characterize ZMYND12 localization and function, we used the *Trypanosoma* model. Forward and reverse genetic approaches are well established for this model, and have largely contributed to the characterization of the molecular pathogenesis of a number of human diseases caused by defective cilia and/or flagella (*Vincensini et al., 2011*). BlastP analysis of the *T. brucei* genome database (*Aslett et al., 2010*) identified the putative ortholog of ZMYND12 as Tb927.9.10370. The *T. brucei* gene encodes TbTAX-1, a 363 amino acid protein previously identified by proteomics analyses as a flagellar protein, annotated as the inner dynein arm protein/p38 (FAP146) (*Broadhead et al., 2006*; *Ralston et al., 2009*). Human ZMYND12 (Q9H0C1) and TbTAX-1 share 29% protein identity and 48% similarity. Both proteins contain a TPR motif domain potentially involved in protein–protein interaction (*D'Andrea and Regan, 2003*; *Figure 3—figure supplement 1*).

RNAi knock-down of TbTAX-1 in the bloodstream form (BSF) of *T. brucei* is reported to lead to multi-flagellated cells, suggesting incomplete cell division (*Broadhead et al., 2006*). This phenotype is described when proteins directly or indirectly involved in flagellar motility are knocked down in the BSF (*Coutton et al., 2018*; *Ralston et al., 2011*). In contrast, in the procyclic form, knock-down induces cell sedimentation due to motility defects (*Baron et al., 2007*).

Here, to analyze the protein's localization by IF on detergent-extracted cells (cytoskeletons, CSK), we generated a procyclic *T. brucei* cell line endogenously expressing myc-tagged TbTAX-1 ($_{myc}$TbTAX-1). $_{Myc}$TbTAX-1 specifically localized to the flagellum, as shown by co-staining with the Paraflagellar Rod-2 protein (PFR2), a marker of the para-axonemal and axoneme-associated structure (PFR) (*Figure 3A*). Importantly, $_{Myc}$TbTAX-1 was found to associate with the axoneme fraction of the flagellum, with staining observed in the distal part of the flagellum, beyond the limit of PFR labeling (*Figure 3A*). Furthermore, $_{myc}$TbTAX-1 labeling was observed in both the old flagellum (OF) and the new flagellum (NF), indicating that the protein is present at all stages of the cell cycle, in line with proteomics data reported for the *Trypanosoma* cell cycle (*Crozier et al., 2018*).

## *T. brucei* TAX-1 is involved in flagellum motility

To assess the functional role of TbTAX-1 in the procyclic parasite model, we used a tetracycline-inducible RNA interference (RNAi) system (*Wirtz et al., 1999*) to knock-down TbTAX-1 expression. Using the $_{myc}$TbTAX-1/TbTAX-1$^{RNAi}$ *T. brucei* cell line (*Figure 3*), we observed the effects of TbTAX-1 knock-down following the addition of tetracycline. Western blot analysis confirmed successful RNA interference, leading to a significant reduction in TbTAX-1 protein levels (*Figure 3B*). These results were confirmed by IF, where expression of $_{Myc}$TbTAX-1 was strongly reduced in the NF compared to the OF (*Figure 3C*). Comparison of TEM images of thin sections from wild-type (WT) and RNAi-induced cells after 7 days revealed no obvious differences in flagellum ultrastructure (*Figure 3—figure supplement 2*), and TbTAX-1 knock-down had no effect on cell growth (*Figure 3D*). However, video microscopy evidence pointed to a strong mobility defect (*Videos 1 and 2*). Tracking analysis (*Figure 3E*) indicated that reduced mobility was linked to problems with flagellar beating. In line with these observations, sedimentation assays showed that after induction for 48 hr, 70% of RNAi-induced cells had sedimented (*Figure 3F*).

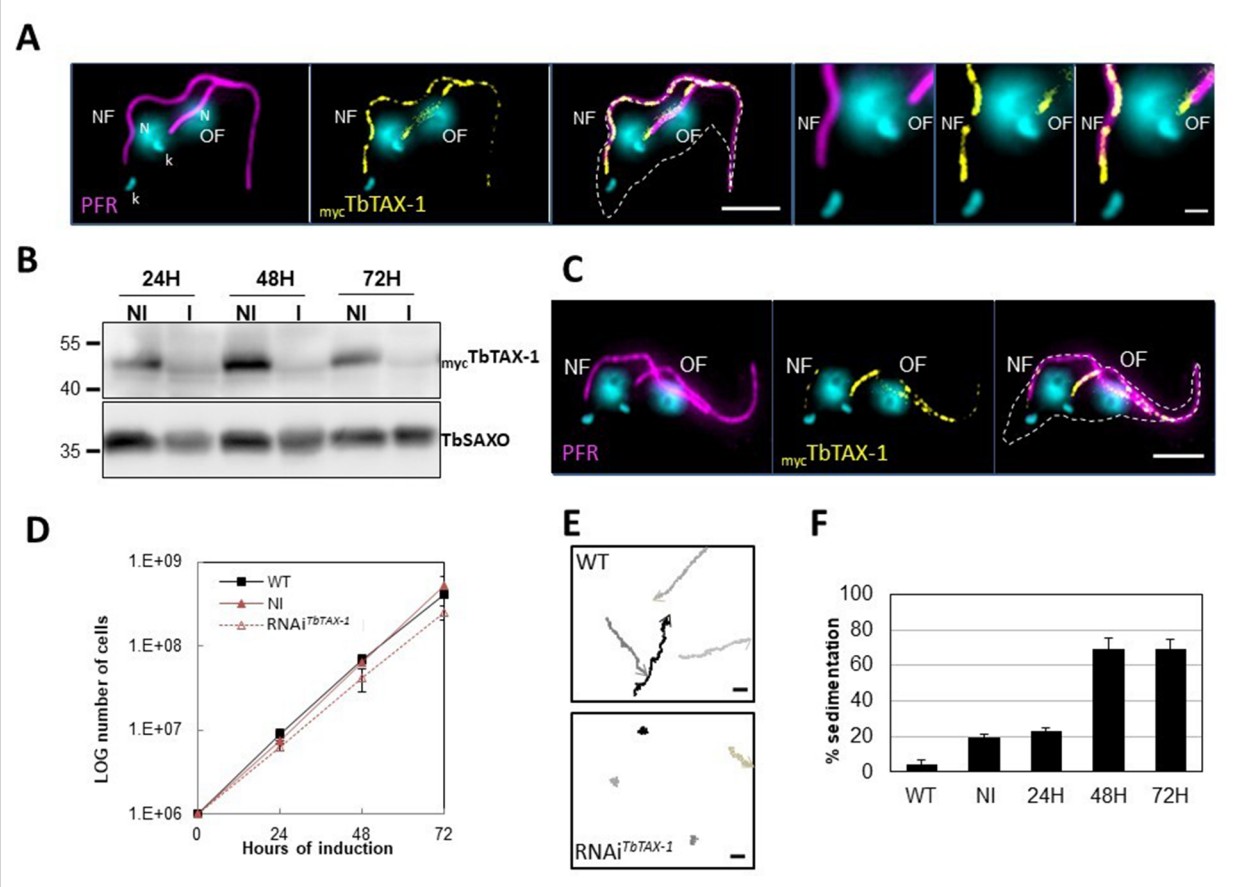

**Figure 3.** TbTAX-1 is an axoneme-associated protein; its knock-down leads to flagellar motility defects. (**A**) Representative image of a post-mitotic detergent-extracted cell immunolabelled with anti-PFR2 (magenta) staining the paraflagellar rod (a para-axonemal structure) and anti-myc (yellow) to reveal ₘᵧᵧTbTAX-1. The protein was detected on the axoneme in both the old flagellum (OF) and the new flagellum (NF). Immunolabeling indicated that the axonemal localization of ₘᵧᵧTbTAX-1 extended distally along the flagellum beyond the region labelled by PFR2 (see zoom). The mitochondrial genome (kinetoplast, k) and nuclei (N) were stained with DAPI. The outline of the cell body is indicated by the dashed white line. Scale bars: 5 μm in whole-cell images; 1 μm in zoomed images. (**B**) Anti-myc western blot analysis of RNAi*ᵀᵇᵀᴬˣ⁻¹* knock-down 24-, 48-, or 72-hr post-induction. ₘᵧᵧTbTAX-1 protein levels were reduced in induced cells (I) compared to non-induced cells (NI); TbSAXO, a flagellum-specific protein, was used as loading control. (**C**) Representative image of a detergent-extracted post-mitotic cell after RNAi*ᵀᵇᵀᴬˣ⁻¹* induction, showing a strong decrease in ₘᵧᵧTbTAX-1 labeling in the NF compared to the OF. (**D**) Comparative *T. brucei* growth curves for wild-type (WT), non-induced (NI), and tetracycline-induced RNAi*ᵀᵇᵀᴬˣ⁻¹* cell lines at 0-, 24-, 48-, and 72-hr post-induction. (**E**) Mobility tracking from video microscopy recordings of live cells: WT and RNAi*ᵀᵇᵀᴬˣ⁻¹* after induction for 72 hr. The positions of individual cells are plotted at 0.28 s intervals; circles indicate start positions, arrowheads indicate end positions. A dramatic loss of progressive mobility was observed after TbTAX1 protein knock-down. Scale bars: 20 μm. (**F**) Sedimentation assays for *T. brucei* WT, non-induced (NI), and tetracycline-induced RNAi*ᵀᵇᵀᴬˣ⁻¹* at 24-, 48-, and 72-hr post-induction. The percentage of cells sedimenting increased from 20 to 70% after depletion of the TbTAX-1 protein.

The online version of this article includes the following source data and figure supplement(s) for figure 3:

**Source data 1.** Annotated and uncropped western blots and raw images for *Figure 3B*.

**Figure supplement 1.** TbTAX-1 is the *T. brucei* ortholog of ZMYND12.

**Figure supplement 2.** TbTAX-1 RNAi knock-down.

## The truncated TbTAX-1 protein is not associated with the flagellum and cannot rescue the motility defect induced by TbTAX-1 RNAi

The p.Arg145Ter mutation of *ZYMND12* identified in MMAF individuals could theoretically produce a truncated form of the protein (aa 1–144). To assess where a similar truncated protein – TbTAX-1 (aa 1–128) – would localize, we generated a TAX-1ₕₐ *Trypanosoma* cell line expressing the truncated form TbTAX-1ᵀʳᵘⁿᶜ_ᵀᵧ₁ alongside the tagged full-length form. The relative sizes of the mutated, truncated ZMYND12 protein potentially encoded in human patients and the equivalent modified,

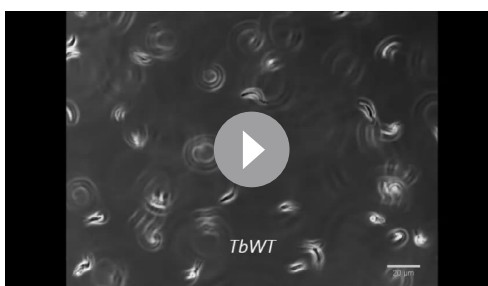

**Video 1.** Video microscopy of wild-type (WT) cells.
https://elifesciences.org/articles/87698/figures#video1

truncated protein – TbTAX1$^{Trunc}$ – are presented in *Figure 3—figure supplement 1B*.

In this cell line, expression of *TbTAX-1$^{RNAi}$* – specifically targeting the full-length protein and not the truncated form – was inducible (*Figure 4A*). Western blot analysis of fractionated samples (whole cells (WC), detergent-extracted cells or CSK, and the flagellum fraction (FG)), we localized the truncated TbTAX-1$^{Trunc}_{Ty1}$ protein (*Figure 4B*). The full-length TbTAX-1 protein was strongly associated with the axoneme, and was detected in the salt-extracted flagellum fraction. In contrast, the truncated TbTAX-1$^{Trunc}_{Ty1}$ form was detected neither in the flagellum nor in the CSK-associated fractions (*Figure 4B*). These results were confirmed by IF on CSK (*Figure 4C*). As expected, induction of *TbTAX-1$^{RNAi}$* led to decreased TbTAX-1 levels, but had no effect on TbTAX-1$^{Trunc}_{Ty1}$ (*Figure 4D*). Cell growth was not affected (*Figure 4E*), and the sedimentation phenotype was still observed (*Figure 4F*). Thus, it appears that the N-terminal domain of TbTAX-1 is not involved in protein targeting and function, and cannot complement TbTAX-1 RNAi knock-down.

## TbTTC29 and TAX-1 belong to the same protein complex; TbTTC29 expression depends on TbTAX-1

As indicated above, in IF analyses of spermatozoa from individuals bearing a mutated ZYMND12, proteins CFAP70, WDR66, TTC29, and SPAG6 were undetected. To further study associations between these proteins, we generated *T. brucei* TbTAX-1$_{HA}$/*TbTAX-1$^{RNAi}$* background cell lines expressing their respective orthologs, that is, TbCFAP70$_{Ty1}$ (Tb927.10.2380), TbWDR66$_{Ty1}$ (Tb927.3.1670), TbTTC29$_{Ty1}$ (Tb927.3.1990), and $_{Ty1}$TbPF16/SPAG6 (Tb927.1.2670).

Western blotting revealed that CFAP70, WDR66, and SPAG6 were tightly associated with the flagellum fraction (*Figure 4—figure supplement 1A*). However, none of these proteins was pulled down with Tb-TAX1 upon co-IP with the anti-Ty1 antibody (*Figure 4—figure supplement 1B*). Furthermore, RNAi knock-down of TbTAX-1 expression did not alter their overall levels (*Figure 4—figure supplement 1C*). The lack of direct interaction between TbTAX-1 and these three proteins was further confirmed by ultrastructure expansion microscopy (U-ExM) IF (*Gambarotto et al., 2019*; *Kalichava and Ochsenreiter, 2021*), which provided no evidence of colocalization (*Figure 4—figure supplement 1D*).

In contrast, decreased TbTAX-1 levels were associated with decreased TbTTC29 expression, and vice versa according to western blot quantification (*Figure 5A and B*). By IF, TbTAX-1 was also observed to co-localize with TbTTC29 on the axoneme (*Figure 5C*). However, analysis of U-ExM and confocal microscopy intensity peaks revealed the colocalization to be incomplete (*Figure 5D*). Nevertheless, IP of TbTTC29$_{Ty1}$ from a TbTTC29$_{Ty1}$/TbTAX-1$_{HA}$ cell line pulled down TbTAX-1$_{HA}$ and vice versa (*Figure 5E*). These results suggest that TbTAX-1 and TbTTC29 are part of the same protein complex.

The co-immunoprecipitated proteins were further analyzed by mass spectrometry. Comparative analysis of the significant hits in each co-IP (anti-Ty1 and anti-HA) (enrichment fold ≥2) identified two additional proteins enriched in both IPs: Tb927.11.3880 – a flagellar actin-like protein (*Ersfeld and Gull, 2001*) – and Tb927.11.8160 – a putative inner arm dynein heavy chain protein belonging to the IAD-4 protein family (data summarized in *Supplementary file 1*). Tb927.11.8160 is potentially involved in flagellum motility (*Wickstead and Gull, 2007*), it localizes along the *T. brucei* axoneme according to the TrypTag genome-wide protein localization resource (http://tryptag.org/; *Dean et al., 2017*).

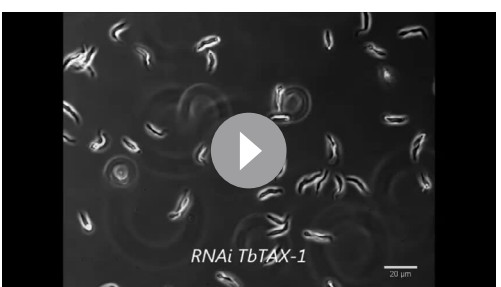

**Video 2.** Video microscopy of RNAi$^{TbTAX-1}$ cells 7 days post-induction.
https://elifesciences.org/articles/87698/figures#video2

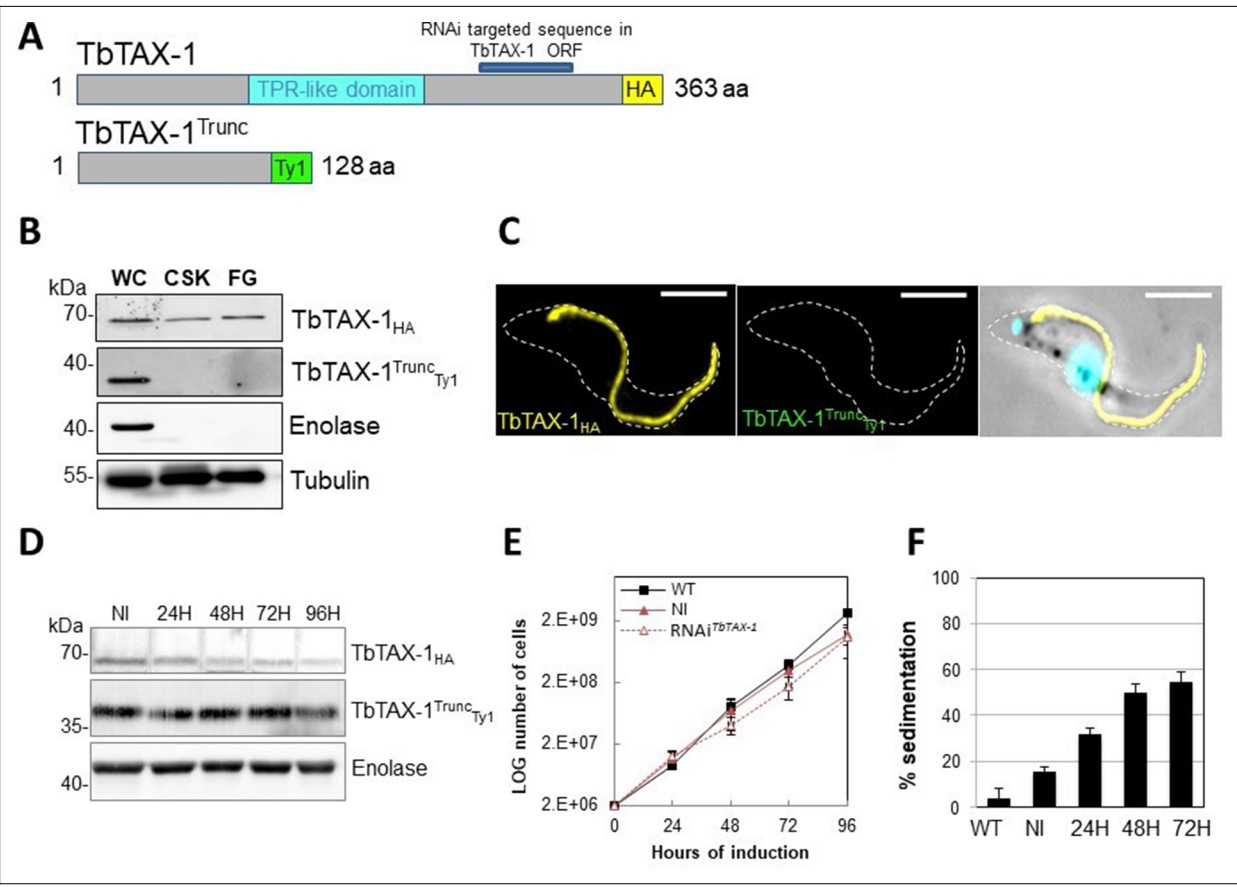

**Figure 4.** The truncated form of TbTAX-1 is not functional. (**A**) Schematic representation of full-length TbTAX-1 and its truncated form. The TPR-like domain is illustrated in light blue, and the targeted RNA interference sequence in dark blue. (**B**) Western blot analysis of subcellular fractions to determine localization of TbTAX-1$_{HA}$ and its truncated form TbTAX-1$^{Trunc}_{Ty1}$. Enolase (cytoplasm) and tubulin (cytoskeleton) were used as loading controls. (**C**) Immunofluorescence labeling of detergent-extracted cytoskeletons to detect TbTAX-1$_{HA}$ (yellow) and TbTAX-1$^{Trunc}_{Ty1}$ (stained green, but not visible as the protein was eliminated during detergent extraction). Scale bars: 5 µm. (**D**) Western blot analysis of the impact of RNAi$^{TbTAX-1}$ expression on TbTAX-1$^{Trunc}_{Ty1}$ levels. Enolase was used as a loading control. (**E**) Growth curves of wild-type (WT), non-induced (NI), and RNAi$^{TbTAX-1}$-induced cells expressing TbTAX-1$_{HA}$ and TbTAX-1$^{Trunc}_{Ty1}$. (**F**) Sedimentation assays for *T. brucei* WT, non-induced (NI), and RNAi$^{TbTAX-1}$-induced cells at 24, 48, and 72 hr.

The online version of this article includes the following source data and figure supplement(s) for figure 4:

**Source data 1.** Annotated and uncropped western blots and raw images for *Figure 4B*.

**Source data 2.** Annotated and uncropped western blots and raw images for *Figure 4D*.

**Figure supplement 1.** TbTAX-1 knock-down does not affect TbWDR66, TbCFAP70, or TbSPAG6.

**Figure supplement 1—source data 1.** Annotated and uncropped western blots and raw images for *Figure 4—figure supplement 1A*.

**Figure supplement 1—source data 2.** Annotated and uncropped western blots and raw images for *Figure 4—figure supplement 1B*.

**Figure supplement 1—source data 3.** Annotated and uncropped western blots and raw images for *Figure 4—figure supplement 1C*.

Significantly, BlastP searches of mammalian genomes for Tb927.11.8160 orthologs identified DNAH1 (Protein ID: XP_828890.1).

## DNAH1 and DNALI1 both interact with TTC29 in mouse

In a recent human study, we showed that the presence of bi-allelic *TTC29* variants induces an MMAF phenotype and male infertility (*Lorès et al., 2019*). By CRISPR/Cas9 mutagenesis and RNAi silencing, we demonstrated that the orthologous TTC29 proteins in mouse and in *T. brucei*, respectively, also constitute flagellar proteins that are required for proper flagellum beating, indicating an evolutionarily conserved function for TTC29 (*Lorès et al., 2019*).

Here, we took advantage of the availability of the mutant mouse model to further decipher the protein complexes associated with TTC29 function within the flagella. We performed comparative

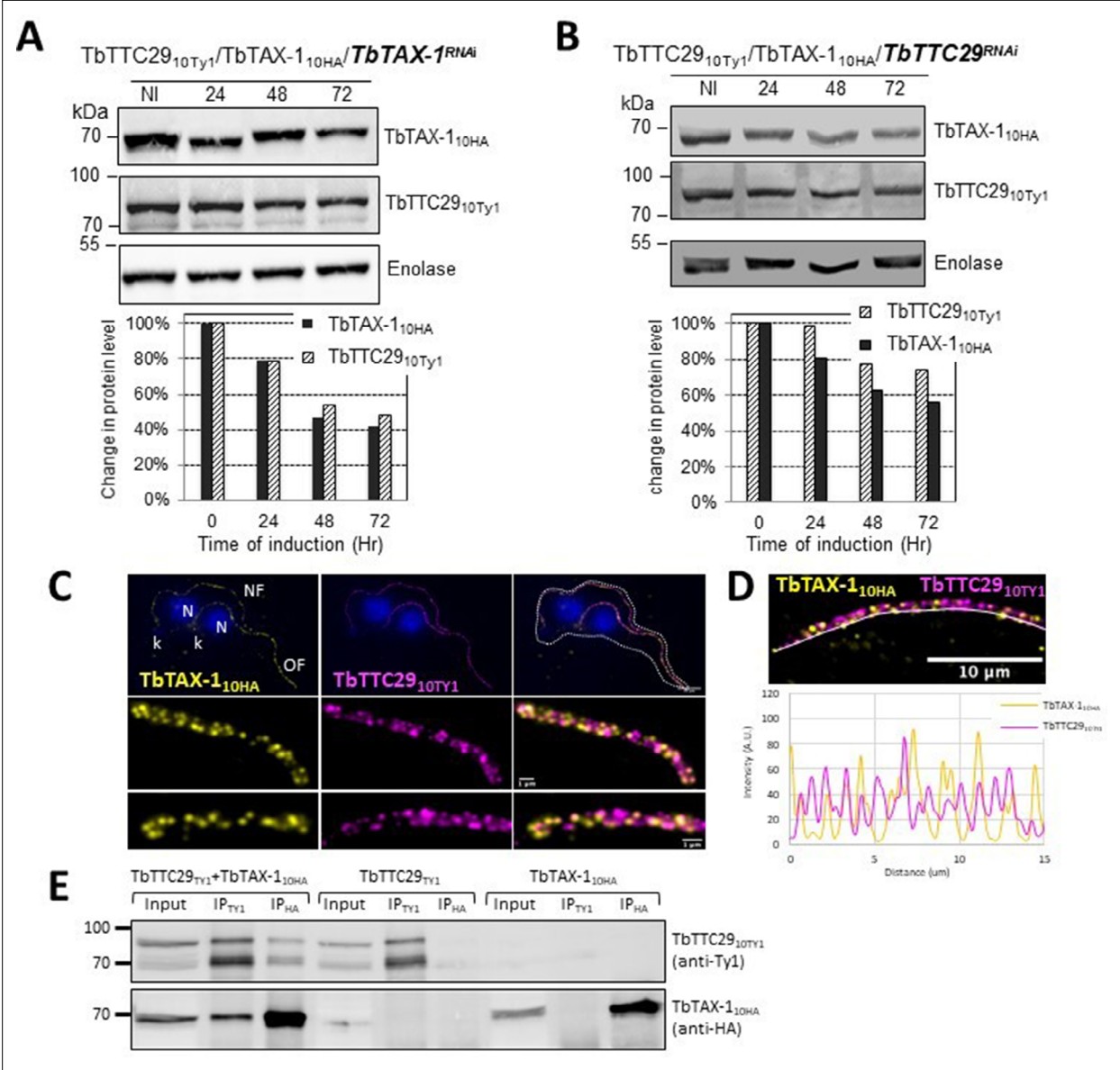

**Figure 5.** TbTAX-1 and TbTTC29 are part of the same protein complex. (**A**) Analysis of the impact of *TbTAX-1* RNAi on TbTTC29$_{Ty1}$ protein levels. Western blot (upper panel) and quantification (lower panel). (**B**) Analysis of the impact of *TbTTC29* RNAi induction on TbTAX-1$_{HA}$ protein levels. Western blot (upper panel) and quantification (lower panel). (**C**) Ultrastructure expansion microscopy (U-ExM) analysis of TbTAX-1$_{HA}$ (yellow) and TbTTC29$_{TY1}$ (purple) colocalization in a dividing cell with an old flagellum (OF) and a new flagellum (NF). Kinetoplast (k) and nuclei (N) were counterstained with Hoechst (blue). (**D**) Confocal image of U-ExM labeling of TbTAX-1$_{HA}$ (yellow) and TbTTC29$_{TY1}$ (magenta) (upper panel). Lower panel: profile plot along the white line showing overlapping intensity peaks for TbTAX-1 and TbTTC29, confirming true colocalization. (**E**) Anti-Ty1 (IP$_{Ty1}$) or anti-HA (IP$_{HA}$) co-immunoprecipitation assays in cell lines expressing both TbTAX-1$_{HA}$ and TbTTC29$_{Ty1}$, or TbTTC29$_{Ty1}$ alone or TbTAX-1$_{HA}$ alone to verify IP specificity. Proteins were detected simultaneously by western blotting using anti-Ty1 and anti-HA antibodies and fluorescent anti-mouse (detecting mouse anti-Ty1) and anti-rabbit (detecting rabbit anti-HA) secondary antibodies. IP assays on cell lines expressing only one or the other tagged protein did not pull down the other tagged protein.

The online version of this article includes the following source data for figure 5:

**Source data 1.** Annotated and uncropped western blots and raw images for *Figure 5A*.

**Source data 2.** Annotated and uncropped western blots and raw images for *Figure 5B*.

**Source data 3.** Annotated and uncropped western blots and raw images for *Figure 5E*.

analyses of co-immunoprecipitations of testis protein extracts from WT mice using TTC29-specific antibodies, and from two previously generated TTC29 KO mouse lines lacking the TTC29 protein (*Ttc29*$^{-/-}$ L5 and L7) (*Lorès et al., 2019*). Mass spectrometry analyses of four independent experimental replicates for each mouse line identified a set of 124 proteins specifically immunoprecipitated in WT testes.

These proteins were further examined to identify TTC29-binding proteins (*Supplementary file 2*). Filtering of immunoprecipitated proteins based on peptide number and replicate constancy identified DNAH1 and DNALI1, in addition to TTC29, as the only proteins that were co-immunoprecipitated with a significant number of peptides in all four experimental replicates. In our previous study, DNAH1 and DNALI1 were detected at equal levels in flagella purified from WT and *Ttc29*$^{-/-}$ KO mice (*Lorès et al., 2019*). Therefore, we considered DNAH1 and DNALI1 to be authentic protein partners of TTC29 in sperm flagella, in line with data obtained with *T. brucei*. Thus, our data indicate that ZMYND12 interacts with DNAH1 and DNALI1, two established components of the axonemal IDA.

## Discussion

The results presented here show that the presence of bi-allelic *ZMYND12* variants induces a typical MMAF phenotype in humans, suggesting that this gene is necessary for sperm flagellum structure and function. ZMYND12 is a previously uncharacterized MYND-domain-containing zinc finger protein, we therefore explored its localization and function using a recognized model: the flagellated protozoan *Trypanosoma*. This model is widely used to investigate flagellum biogenesis and the function of flagellar proteins. In addition to having a very similar axonemal organization and composition to human flagellum, *Trypanosoma* has numerous advantages for functional studies, and is very useful for reverse genetics and biochemical or proteomics studies (*Vincensini et al., 2011*).

We first confirmed that ZMYND12 is an axonemal protein, and then used RNAi experiments to confirm its critical role in flagellum function. ZMYND12 knock-down led to dramatic flagellar motility defects like those observed in human sperm (*Figure 3*). In addition, a truncated form of TAX-1 protein lacking the TPR domain – mimicking the expected truncated ZMYND12 protein in MMAF individuals bearing the p.Arg145Ter variant – was not associated with the flagellum, and failed to rescue TAX-1 RNAi-induced motility defects (*Figure 4*). These results clearly indicate that the TPR motif of ZMYND12 is critical for protein targeting and function in both human sperm and *Trypanosoma*. Other groups also previously reported a significant role for TPR proteins in sperm flagellar biogenesis and male infertility (*Liu et al., 2019b*; *Lorès et al., 2019*). Indeed, TPR domains are known to be involved in protein-protein interaction and the assembly of multiprotein complexes (*D'Andrea and Regan, 2003*). We therefore decided to further explore the potential interactions between ZMYND12 and other axonemal proteins.

IF experiments in sperm cells from ZMYND12 individuals revealed severe ultrastructural defects dramatically affecting several components of the axoneme. More specifically, IDAs and the CSC were severely disorganized, as shown by the total absence of DNAH1, WDR66, and DNALI1 immunostaining (*Figure 2A, B* and *Figure 2—figure supplement 1*). Intriguingly, TTC29 and CFAP70 were also absent from sperm cells from individuals bearing ZMYND12 mutations (*Figure 2C, D*). These observations suggest some functional or physical interaction between these proteins. However, the exact function and location of TTC29 and CFAP70 remain unclear. In *Chlamydomonas*, TTC29 has been suggested to be a component of IDAs (*Yamamoto et al., 2008*), but this protein was also reported to possibly contribute to the IFT machinery (*Brooks, 2014*; *Lorès et al., 2019*). For CFAP70, functional studies in *Chlamydomonas* demonstrated that the protein localized at the base of the ODA, where it could regulate ODA activity (*Shamoto et al., 2018*). In *T. brucei*, we screened the axonemal proteins WDR66, CFAP70, SPAG6, and TTC29 to test their interaction with the ZMYND12 homolog. Our results indicated that TbTAX-1 only interacts with TbTTC29, and that the two proteins are part of the same axonemal complex (*Figure 5A, B*). This interaction was confirmed by U-ExM and confocal microscopy (*Figure 5C, D*). A third member of the complex – Tb927.11.8160, the ortholog of human DNAH1 – was subsequently identified by comparative proteomics analysis of co-immunoprecipitated proteins (*Figure 5E* and *Supplementary file 1*). These results were further confirmed using the *Ttc29*-null mouse model previously generated in our laboratory (*Supplementary file 2*). All of these results are consistent with previous studies in *Chlamydomonas* suggesting that p38 (ZMYND12 ortholog) and p44 (TTC29 ortholog) interact and play a role in IDAd docking on the axoneme (*Yamamoto*

*et al., 2008*). Interestingly, the axonemal IDAd in *Chlamydomona*s is composed of the DHC2 heavy chain and the p28 light chain, orthologs of human DNAH1 and DNALI1, respectively. Moreover, the recent human reference interactome map obtained by high-throughput two-hybrid screening (*Luck et al., 2020*) also supports direct protein–protein interaction between ZMYND12 and TTC29. Our results thus formally demonstrate that ZMYND12 interacts with TTC29 within the axonemal complex, in association with DNAH1 and inner dynein arms. DNAH1 is directly connected through an arc-like structure (*King, 2013*) to the base of radial spoke 3 (RS3), which is anchored to the axoneme by the CSC (*Dymek et al., 2011*). In humans and *Trypanosoma*, the absence of ZMYND12 in the flagellum probably destabilizes this axonemal complex, and consequently alters the stability of the RS3 base-docked IDAs including DNAH1, the CSC, and RS. Each of these defects leads to severe axonemal disorganization, and an MMAF sperm phenotype. Our team previously reported deleterious variants of *TTC29* and *DNAH1* associated with an MMAF phenotype. The flagellar defects caused by these variants are similar to those observed in sperm from individuals with *ZMYND12* defects (*Ben Khelifa et al., 2014*; *Liu et al., 2019a*; *Lorès et al., 2019*). The results presented here thus contribute to elucidating the physical and functional associations between ZMYND12, TTC29, and DNAH1 in the axoneme, and emphasize the key role played by this triumvirate in maintaining the structure and function of the sperm flagellum.

Adding the three ZMYND12-mutated individuals to the 83 individuals previously identified with variants in known MMAF-related genes in our cohort of 167 individuals, we now have a diagnostic efficiency of 51.4% (86/167). This figure demonstrates the efficiency and the clinical utility of WES for the investigation of the genetic causes and pathophysiology of MMAF syndrome. Among the variants identified, we found the same *ZMYND12* nonsense variant c.433C>T; p.(Arg145Ter) in three unrelated individuals from diverse geographical origins (i.e., Tunisia, China), suggesting that the Arg145 residue may constitute a mutational hotspot. Importantly, as previously reported (*Kherraf et al., 2018*), we confirmed that WES can detect small pathogenic exon deletions. These events may constitute a recurrent disease mechanism in MMAF, and should therefore be systematically explored. Despite these significant advances, the genetic cause of MMAF remains unknown for about half the individuals affected, due to the high genetic heterogeneity of the disorder. To improve the diagnostic rate, more powerful techniques such as whole-genome sequencing may now be envisaged for MMAF patients for whom WES fails to provide results (*Meienberg et al., 2016*).

The results presented here demonstrate that the combined use of exome sequencing and the *Trypanosoma* model can efficiently reveal new genes responsible for an MMAF phenotype in human and decipher how they affect sperm flagellum architecture and function. Overall, our strategy identified bi-allelic variants in *ZMYND12* associated with severe flagellum malformations resulting in asthenoteratozoospermia and primary male infertility. In conclusion, our work with human sperm and the flagellated protist *T. brucei* demonstrated that ZMYND12 is part of the same axonemal complex as TTC29 and DNAH1, and plays a critical role in flagellum function and assembly.

## Materials and methods
### Study participants

WES data from a previously established cohort of 167 MMAF individuals were analyzed (*Coutton et al., 2019*). All individuals presented a typical MMAF phenotype characterized by severe asthenozoospermia (total sperm motility below 10%), with >5% of the spermatozoa displaying at least three of the following flagellar abnormalities: short, absent, coiled, bent, or irregular flagella (*Coutton et al., 2019*). All individuals had a normal somatic karyotype (46,XY) with normal bilateral testicular size, hormone levels, and secondary sexual characteristics. Sperm were analyzed in the source laboratories during routine biological examination of the individuals according to World Health Organization (WHO) guidelines (*Wang et al., 2014*). An additional Chinese individual was included in the study from a familial case. This individual was enrolled from the Human Sperm Bank of West China Second University Hospital of Sichuan University; he presented with severe asthenoteratozoospermia and a typical MMAF phenotype. WES analysis for this subject was performed as described in the previous report (*Liu et al., 2021*). Sperm morphology was assessed following Papanicolaou staining (*Figure 1A*). Detailed semen parameters of the four individuals carrying *ZMYND12* mutations (*ZMYND12*$_{1–4}$) are presented in *Table 1*. Sperm samples for additional phenotypic characterization were obtained from

ZMYND12_3. Written informed consent was obtained from all individuals participating in the study, and institutional approval was given by the local medical ethics committee (CHU Grenoble Alpes institutional review board #6705). Samples were stored in the Fertithèque collection registered with the French Ministry of Health (DC-2015-2580) and the French Data Protection Authority (DR-2016-392).

## WES and bioinformatics analysis

Genomic DNA was isolated from EDTA blood using DNeasy Blood & Tissue Kits (QIAGEN SA). Genetic data were obtained through several sequencing centers, in particular Novogene, Genoscope, and Integragen. Coding regions and intron/exon boundaries were sequenced after enrichment using Agilent SureSelect Human All Exon V5, V6, or Clinical Research capture kits. An alignment-ready GRCh38 reference genome (including ALT, decoy, and HLA) was produced using 'run-gen-ref hs38DH' from Heng Li's bwakit package (https://github.com/lh3/bwa; *Li, 2017*). Exomes were analyzed using a bioinformatics pipeline developed in-house. The pipeline consists of two modules, both distributed under the GNU General Public License v3.0 and available on github (grexome-TIMC-Primary TIMC primary, https://github.com/ntm/grexome-TIMC-Primary; copy archived at *Thierry-Mieg, 2023a*, TIMC secondary, https://github.com/ntm/grexome-TIMC-Secondary; copy archived at *Thierry-Mieg, 2023b*). The first module takes FASTQ files as input and produces a single merged GVCF file per variant-caller, as follows: trim adaptors and filter low-quality reads with fastp 0.23.2, align reads with BWA-MEM 0.7.17, mark duplicates using samblaster 0.1.26, and sort and index BAM files with samtools 1.15.1. SNVs and short indels are called from each BAM file using strelka 2.9.10 (*Kim et al., 2018*) and GATK 4.1.8.1 to produce individual GVCF files from each variant-caller. These files are finally merged using mergeGVCFs.pl to obtain a single multi-sample GVCF per caller. Using several variant-callers helps to compensate for each caller's flaws. The second module takes each merged GVCF as input and produces annotated analysis-ready TSV files. Annotation involved 17 streamlined tasks: Low-quality variant calls (DP <10, GQ <20, or less than 15% of reads supporting the ALT allele) were discarded. Variant Effect Predictor v108 (*McLaren et al., 2016*) was used to annotate variants and predict their impact, in order to filter out low-impact (MODIFIER) variants and/or prioritize high-impact ones (e.g., stop-gain or frameshift variants). Variants with a minor allele frequency greater than 1% in gnomAD v.2.1 or 3% in 1000 Genomes Project phase 3 were excluded. The pipeline performs an integrative analysis of all available exomes, both at the individual variant level and at the gene level. Thus, recurring genomic variants are readily identified alongside genes that are severely impacted in several patients but not in control subjects. Additional information can be found in *Arafah et al., 2021*. Copy number variants were searched using the ExomeDepth software package, as previously reported (*Kherraf et al., 2018*; *Plagnol et al., 2012*).

## Sanger sequencing

ZMYND12 single-nucleotide variants identified by exome sequencing were validated by Sanger sequencing as previously described (*Coutton et al., 2019*). PCR primers and protocols used for each individual are listed in *Supplementary file 3*.

## MLPA analysis

Deletion of exons 6–8 identified in individual ZMYND12_2 by exome sequencing was confirmed by custom MLPA analysis, according to our previously described method (*Coutton et al., 2012*). The MLPA probes were designed, and the MLPA reaction was performed according to the recommendations set out in the MRC-Holland synthetic protocol (https://mlpa.com/). Data analysis was also performed in line with this protocol. For this study, three synthetic MLPA probes specific for exons 5, 6, and 8 of ZMYND12 were designed. MLPA probes are listed in *Supplementary file 4*.

## RT-qPCR analysis

RT-qPCR experiments were performed as previously described (*Coutton et al., 2019*) with cDNAs extracted from various human tissues purchased from Life Technologies. A panel of 10 organs was used for experiments: testis, brain, trachea, lung, kidney, liver, skeletal muscle, pancreas, placenta, and heart. Human RT-qPCR data were normalized relative to the two reference housekeeping genes RPL6 and RPL27 by the $-\Delta\Delta Ct$ method (*Livak and Schmittgen, 2001*). Primer sequences and RT-qPCR conditions are indicated in *Supplementary file 5*. Relative expression of ZMYND12 transcripts was

compared in several organs, and statistical analysis involved a two-tailed *t*-test (Prism 4.0 software; GraphPad, San Diego, CA). A p-value ≤0.05 was considered significant.

## Immunostaining in human sperm cells

IF experiments were performed on sperm cells from control individuals, from individual ZMYND12$_{-3}$ – carrier of the nonsense variant c.103C>T – and from a previously reported MMAF individual – carrier of the *TTC29* splicing variant c.1761+1G>A (*Lorès et al., 2019*). For each MMAF individual, 200 sperm cells were analyzed by two experienced operators, and the IF staining intensity and patterns were compared to those obtained from fertile control individuals. Sperm cells were fixed in phosphate-buffered saline (PBS)/4% paraformaldehyde for 1 min at room temperature. After washing in 1 ml PBS, the sperm suspension was spotted onto 0.1% poly L-lysine pre-coated slides (Thermo Scientific). After attachment, sperm were permeabilized with 0.1% (vol/vol) Triton X-100–DPBS (Dulbecco's Phosphate-buffered saline) (Triton X-100; Sigma-Aldrich) for 5 min at RT. Slides were then blocked by applying 5% normal serum–DPBS (normal goat or donkey serum; Gibco, Invitrogen) before incubating overnight at 4°C with the primary antibody. The following primary antibodies were used: ZMYND12, DNAI1, DNALI1, DNAH1, DNAH8, DNAH17, RSPH1, SPAG6, GAS8, CFAP70, WDR66 (also known as CFAP251), TTC29, AKAP4, and acetylated-α-tubulin. The primary antibody references and dilutions are fully detailed in *Supplementary file 6*. Slides were washed with 0.1% (vol/vol) Tween 20–DPBS before incubating for 1 hr at room temperature with secondary antibodies. Highly cross-adsorbed secondary antibodies (Dylight 488 and Dylight 549, 1:1000) were purchased from Jackson Immuneresearch. Experimental controls were treated similarly, but omitting the primary antibodies. Samples were counterstained with DAPI and mounted with DAKO mounting media (Life Technology). Fluorescence images were captured with a confocal microscope (Zeiss LSM 710).

## *T. brucei* cell lines, culture, and transfection

The trypanosome cell lines used in this study are derived from the procyclic parental form, *T. brucei* SmOxP427 (*Sunter, 2016*). The absence of contamination of the cell line by Mycoplasma was verified by PCR. Our PCR and sequencing data also confirmed that all cell lines were *T. brucei brucei* Lister 427 and carried in situ or episomal the designated sequence. These cells co-express the T7 RNA polymerase and the tetracycline repressor and were grown and transfected as previously described (*Kherraf et al., 2018*). When required, the culture medium was supplemented with puromycin (1 µg/ml), neomycin (10 µg/ml), phleomycin (5 µg/ml), or hygromycin (25 µg/ml). Cells were transfected using an AMAXA electroporator (Lonza) as previously described (*Schumann Burkard et al., 2011*). *TbTAX-1* RNA interference was induced with tetracycline (10 µg/ml).

Tagged proteins were obtained from SmOxP427 cells, by inserting epitope-tagged transgenes at the endogenous locus. Transgenes were N- or C-terminally tagged with 10myc, 10Ty1, or 10HA tags using the pPOTv7 vector series, as previously described (*Dean et al., 2015*; *Kherraf et al., 2018*). For RNAi *TbTAX-1* knock-down, bp 219–667 were cloned between the *XhoI* and *XbaI* sites of p2T7tiB (*LaCount et al., 2002*) and the linearized vector was transfected into endogenously tagged TbTAX-1-background cell lines. To target the full-length *TbTAX-1* and not the shorter form, similar cloning was performed with bp 493–917. The TbTTC29$_{Ty1}$ RNAi cell line is described elsewhere (*Lorès et al., 2019*).

## Co-immunoprecipitation and mass spectrometry on *Trypanosoma* cells

In a 15-ml polycarbonate tube, $2 \times 10^8$ cells expressing the different tagged proteins were washed in PBS and incubated for 5 min at RT in 1 ml of immunoprecipitation buffer for cell lysis (IP buffer; 25 mM Tris–HCl pH 7.4, 100 mM NaCl, 0.25% Nonidet P-40 (Igepal), 1 mM DTT(Dithiothréitol), protease inhibitor cocktail, and 1 unit of nuclease) (Pierce). Lysis was completed by sonication (5 × 30 s ON/30 s OFF) in a Bioruptor sonication system. Cell lysates were clarified by centrifugation (10 min, 16,000 × *g*, 4°C) and split into two samples each. One was incubated with 3 µl of anti-Ty1 and the other with 2 µl of mouse anti-HA for 1 hr at 4°C on a rotating wheel. Protein-G Dynabead (Invitrogen 10003D) slurry (40 µl) was washed in IP buffer and incubated with the samples for 1 hr at 4. After washing three times in IP buffer, beads were resuspended in 40 µl sample buffer and boiled before loading 20 µl on sodium dodecyl sulfate–polyacrylamide gel electrophoresis (SDS–PAGE).

For mass spectrometry analyses, $1.8 \times 10^9$ cells were washed in PBS, resuspended in 2 ml of immunoprecipitation buffer without DTT and processed as described above. Antibodies used were: mouse anti-HA (10 µl) and anti-TY1 (15 µl). Complex precipitation was achieved by adding 200 µl of Protein-G Dynabead slurry. After washing three times in IP buffer, beads were transferred to new tubes and washed 3× in PBS before storing dry at −20°C.

Protein-G Dynabeads were resuspended in Laemmli buffer and boiled. Supernatants were loaded on a 10% acrylamide SDS–PAGE gel, and proteins were revealed by Colloidal Blue staining. Sample preparation and protein digestion by trypsin were performed as previously described (*Allmann et al., 2014*). NanoLC-tandem mass spectrometry (MS/MS) analyses were performed using an Ultimate 3000 RSLC Nano-UPHLC system (Thermo Scientific, USA) coupled to a nanospray Orbitrap Fusion Lumos Tribrid Mass Spectrometer (Thermo Fisher Scientific, California, USA). Each peptide extract was loaded on a 300 µm ID × 5 mm PepMap C18 pre-column (Thermo Scientific, USA) at a flow rate of 10 µl/min. After a 3-min desalting step, peptides were separated on a 50-cm EasySpray column (75 µm ID, 2 µm C18 beads, 100 Å pore size; ES903, Thermo Fisher Scientific) by applying a 4–40% linear gradient of solvent B (0.1% formic acid in 80% acetonitrile [ACN]) over 48 min. The flow rate for separation was set to 300 nl/min. The mass spectrometer was operated in positive ion mode at a 2.0 kV needle voltage. Data were acquired using Xcalibur 4.4 software in data-dependent mode. MS scans (*m/z* 375–1500) were recorded at a resolution of $R = 120,000$ (@ *m/z* 200) and an Automated Gain Control (AGC) target of $4 \times 10^5$ ions collected within 50 ms was applied along with a top speed duty cycle of up to 3 s for MS/MS acquisition. Precursor ions (2–7 charge states) were isolated in the quadrupole within a mass window of 1.6 Th, and fragmented by Higher-energy Collisional Dissociation (HCD), applying 28% normalized collision energy. MS/MS data were acquired in the Orbitrap cell at a resolution of $R = 30,000$ (*m/z* 200), a standard AGC target and a maximum injection time in automatic mode. A 60-s exclusion window was applied to previously selected precursors. Proteome Discoverer 2.5 was used to identify proteins and perform label-free quantification (LFQ). Proteins were identified in batch mode using MS Amanda 2.0, Sequest HT, and Mascot 2.5 algorithms, with searches performed against a *T. brucei brucei* TREU927 protein database (9788 entries, release 57, https://tritrypdb.org/ website). Up to two missed trypsin cleavages were allowed. Mass tolerances in MS and MS/MS were set to 10 ppm and 0.02 Da, respectively. Oxidation (M) and acetylation (K) were included as dynamic modifications, and carbamidomethylation (C) as a static modification. Peptides were validated using the Percolator algorithm (*Käll et al., 2007*), retaining only 'high confidence' peptides, corresponding to a 1% false discovery rate at the peptide level. The minora feature-detector node (LFQ) was used along with the feature-mapper and precursor ion quantifier. The following quantification parameters were applied: (1) unique peptides; (2) precursor abundance based on intensity; (3) no normalization; (4) protein abundance calculation: summed abundances; (5) protein ratio calculation: pairwise ratios; (6) imputation mode: low abundance resampling; and (7) hypothesis test: *t*-test (background based). For master proteins, quantitative data were considered based on a minimum of two unique peptides, a fold-change greater than 2 and a Benjamini–Hochberg-corrected p-value for the FDR of less than 0.05. Mass spectrometry proteomics data have been deposited to the ProteomeXchange Consortium via the PRIDE partner repository (*Adhikari et al., 2020*), under dataset identifier PXD039470.

## IF on *Trypanosoma* cells

Cells were harvested, washed, and processed for immunolabeling on methanol-fixed detergent-extracted cells (CSK) as described in *Albisetti et al., 2017*. The antibodies used and their dilutions are listed in *Supplementary file 7*. Images were acquired on a Zeiss Imager Z1 microscope, using a Photometrics Coolsnap HQ2 camera, with a ×100 Zeiss objective (NA 1.4) using Metamorph software (Molecular Devices), and processed with ImageJ (NIH).

## U-ExM on *Trypanosoma* cells

The U-ExM protocol has been adapted and optimized for use on trypanosomes, as described (*Gambarotto et al., 2021*). Protocol details can be found online at https://doi.org/10.17504/protocols.io.bvwqn7dw (*Casas et al., 2022*). Images were acquired on a Zeiss AxioImager Z1 using a ×63 (1.4 NA) objective, or at the Bordeaux Imaging Center with a Leica TCS SP5 on an upright stand DM6000 (Leica Microsystems, Mannheim, Germany), fitted with a HCX Plan Apo CS ×63 oil NA 1.40

objective. Images were processed and analyzed using ImageJ and Fiji software (*Schindelin et al., 2012*; *Schneider et al., 2012*).

## *Trypanosoma* cells: sedimentation assays and video microscopy

Sedimentation assays were performed as described (*Dacheux et al., 2012*). Briefly, cells were placed in cuvettes and incubated for 24 hr without shaking. Optical density ($OD_{600\,nm}$) was measured before mixing (ODb, to detect 'swimming' cells) and after mixing (ODa, to detect 'swimming' plus 'sedimenting' cells). The percentage of sedimented cells was calculated as $100 - (ODb/ODa) \times 100$ for the WT and RNAi cells. Sedimentation of RNAi cells was normalized relative to the tagged TbTAX-1 parental cell line. Video microscopy was carried out as described (*Lorès et al., 2019*). Briefly, PCF(Procyclic Form) cells (WT cell and RNAi*TbTAX*-1 cells 72 hr after induction) were washed in PBS, and cell mobility was recorded by phase-contrast microscopy on a Zeiss AxioImager Z1, ×40 lens (NA 1.3). A 28-s sequence of digital video was captured from separate regions and analyzed using Metamorph software (Molecular Devices) (speed x8). The positions of individual cells were plotted over 0.28-s intervals. The start and end positions of each cell were marked.

## TEM on *Trypanosoma* cells

TEM blocks of *Trypanosoma* cells and thin sections were prepared as previously described (*Albisetti et al., 2017*). Thin sections were visualized on an FEI Tecnai 12 electron microscope. Images were captured on an ORIUS 1000 11M Pixel camera (resolution 3 ± 5 nm) controlled by Digitalmicrograph, and processed using ImageJ (NIH).

## Western blot analyses of *Trypanosoma* cells

Proteins from WCs, detergent-extracted CSK and flagella, or co-IPs were separated on SDS–PAGE gels. Separated proteins were transferred by semi-dry transfer to PVDF(polyvinylidene difluoride) and low-fluorescence PVDF membranes for further processing, as described (*Albisetti et al., 2017*). The antibodies used and their dilutions are listed in *Supplementary file 7*. For IP analysis, primary antibodies – anti-TY1 and rabbit anti-HA (GTX115044) – and fluorescent secondary antibodies – anti-mouse Starbright Blue 520 (Bio-Rad 12005867) and anti-rabbit Starbright Blue 700 (Bio-Rad 12004162) – were used.

## Co-immunoprecipitation and mass spectrometry analysis on $Ttc29^{-/-}$ mouse testes

Testes from WT and $Ttc29^{-/-}$ L5 and L7 mice (*Lorès et al., 2019*) were dissected in cold PBS; the albuginea was removed and the seminiferous tubules were dissociated and pelleted by brief centrifugation. Proteins from the pellets were extracted in 1 ml lysis buffer: 50 mM Tris–HCl, pH 8, 150 mM NaCl, 10 mM $MgCl_2$, 0.5% NP40, and Complete protease inhibitor cocktail (Roche Applied science). Immunoprecipitation was performed on total-protein lysate using 1 µg of rabbit polyclonal TTC29 antibody (HPA061473, Sigma-Aldrich) and 15 µl of Protein A/G magnetics beads (Bio-Ademtec) according to the manufacturer's instructions. The experiment was replicated with four mice for each genotype.

Total immunoprecipitated proteins were recovered in 25 µl 100 mM Tris/HCl pH 8.5 containing 10 mM TCEP(tris(2-carboxyéthyl)phosphine), 40 mM chloroacetamide, and 1% sodium deoxycholate, by sonicating three times, and incubating for 5 min at 95°C. An aliquot (20 µg) of protein lysate was diluted (1:1) in Tris 25 mM pH 8.5 in 10% ACN and subjected to overnight trypsin digestion with 0.4 µg of sequencing-grade bovine trypsin (Promega) at 37°C. Deoxycholate was removed using liquid–liquid phase extraction after adding 50 µl of 1% trifluoroacetic acid (TFA) in ethyl-acetate. A six-layer home-made Stagetip with Empore Styrene- divinylbenzene – Reversed Phase Sulfonate (SDB RPS; 3M) disks was used. Eluted peptides were dried and re-solubilized in 2% TFA before preparing five fractions on hand-made Strong Cation eXchange (SCX) StageTips (*Kulak et al., 2014*). Fractions were analyzed on a system combining an U3000 RSLC nanochromatograph eluting into an Orbitrap Fusion mass spectrometer (both Thermo Scientific). Briefly, peptides from each SCX fraction were separated on a C18 reverse phase column (2 µm particle size, 100 Å pore size, 75 µm inner diameter, 25 cm length) with a 145-min linear gradient starting from 99% solvent A (0.1% formic acid in $H_2O$) and ending with 40% of solvent B (80% ACN and 0.085% formic acid in $H_2O$). The MS1 scans spanned 350–1500 *m/z*

with an AGC target of $1 \times 10^6$ within a maximum ion injection time (MIIT) of 60 ms, at a resolution of 60,000. The precursor selection window was set to 1.6 *m/z* with quadrupole filtering. HCD Normalized Collision Energy was set to 30%, and the ion trap scan rate was set to 'rapid' mode with AGC target $1 \times 10^5$ and 60 ms MIIT. Mass spectrometry data were analyzed using Maxquant (v.1.6.1.0) (*Cox et al., 2014*). The database used was a concatenation of human sequences from the Uniprot-Swissprot database (release 2017-05) and the Maxquant list of contaminant sequences. Cysteine carbamidomethylation was set as fixed modification, and acetylation of protein N-terminus and oxidation of methionine were set as variable modifications. Second peptide search and 'match between runs' (MBR) options were allowed. Label-free protein quantification (LFQ) was performed using both unique and razor peptides, requiring at least two peptides per protein. The quality of raw quantitative data was evaluated using PTXQC software v. 0.92.3 (*Bielow et al., 2016*). Perseus software, version 1.6.2.3 (*Tyanova et al., 2016*) was used for statistical analysis and data comparison. Proteomics data obtained by mass spectrometry analysis were deposited to the ProteomeXchange Consortium via the PRIDE partner repository (*Adhikari et al., 2020*) under dataset identifier PXD039854.

## Acknowledgements

We thank all individuals for their participation. We thank Samuel Dean (Warwick Medical School) and Jack Sunter (Oxford Brookes University) for the pPOT plasmids and SmOx T brucei cell lines, F Bringaud (University of Bordeaux) for the anti-enolase antibody, and P Bastin (Institut Pasteur) for the anti-Ty1 antibody. We thank G Cougnet-Houlery, S Guit, and J Marcos for ongoing access to the MFP lab infrastructure. This work was supported by the Institut National de la Santé et de la Recherche Médicale (INSERM), the Centre National de la Recherche Scientifique (CNRS), University Grenoble Alpes, and French National Research Agency funding for specific projects: MASFLAGELLA (ANR-14-CE15-0002), FLAGEL-OME (ANR-19-CE17-0014), and OLIGO-SPERM (ANR-21-CE17-0007). Mass spectrometry experiments were performed at the Centre de Génomique Fonctionnelle facility (CGFB, Bordeaux) for *Trypanosoma* studies and at the Proteom'IC core-facility (Institut Cochin, Paris – François Guillonneau and Johanna Bruce) for mouse studies. The Orbitrap Fusion mass spectrometer at the Proteom'IC core-facility was acquired with funds from the FEDER through the « Operational Program for Competitiveness Factors and employment 2007–2013 » and from the « Cancéropôle Ile-de-France ». The Bordeaux Imaging Center is a service unit of the CNRS INSERM and Bordeaux University, a member of the French BioImaging national infrastructure funded through the French National Research Agency [ANR-10-INBS-04].

## Additional information

### Funding

| Funder | Grant reference number | Author |
|---|---|---|
| Agence Nationale de la Recherche | ANR-19-CE17-0014 | Pierre F Ray |
| Agence Nationale de la Recherche | ANR-21-CE17-0007 | Charles Coutton |
| Agence Nationale de la Recherche | ANR-10-INBS-04 | Mélanie Bonhivers |

The funders had no role in study design, data collection, and interpretation, or the decision to submit the work for publication.

### Author contributions

Denis Dacheux, Resources, Data curation, Formal analysis, Funding acquisition, Validation, Methodology, Writing – original draft, Writing – review and editing; Guillaume Martinez, Conceptualization, Data curation, Formal analysis, Methodology, Writing – original draft, Writing – review and editing; Christine E Broster Reix, Formal analysis, Methodology, Writing – original draft; Julie Beurois, Resources, Formal analysis, Investigation, Writing – original draft; Patrick Lores, Data curation, Formal

analysis, Writing – original draft, Writing – review and editing; Magamba Tounkara, Resources, Formal analysis, Validation, Investigation, Methodology, Writing – original draft; Jean-William Dupuy, Data curation, Formal analysis; Derrick Roy Robinson, Conceptualization, Formal analysis, Methodology, Writing – original draft, Writing – review and editing; Corinne Loeuillet, Resources, Formal analysis, Methodology; Emeline Lambert, Formal analysis, Methodology; Zeina Wehbe, Resources, Formal analysis; Jessica Escoffier, Resources, Data curation, Formal analysis, Funding acquisition; Amir Amiri-Yekta, Resources, Formal analysis, Investigation, Project administration; Abbas Daneshipour, Formal analysis, Supervision, Investigation, Methodology; Seyedeh-Hanieh Hosseini, Formal analysis; Raoudha Zouari, Xiaohui Jiang, Formal analysis, Funding acquisition, Investigation, Methodology; Selima Fourati Ben Mustapha, Formal analysis, Investigation; Lazhar Halouani, Formal analysis, Investigation, Methodology; Ying Shen, Formal analysis, Investigation, Methodology, Writing – original draft; Chunyu Liu, Formal analysis, Investigation, Methodology, Writing – original draft, Project administration; Nicolas Thierry-Mieg, Conceptualization, Data curation, Formal analysis, Investigation, Methodology, Writing – original draft, Project administration; Amandine Septier, Software, Formal analysis; Marie Bidart, Formal analysis, Funding acquisition, Methodology, Writing – original draft; Véronique Satre, Data curation, Formal analysis, Funding acquisition, Validation, Methodology, Writing – original draft; Caroline Cazin, Resources, Data curation, Formal analysis, Methodology, Writing – original draft; Zine Eddine Kherraf, Data curation, Formal analysis, Validation, Investigation, Methodology, Writing – original draft, Project administration; Christophe Arnoult, Conceptualization, Data curation, Supervision, Funding acquisition, Validation, Investigation, Methodology, Writing – original draft, Project administration; Pierre F Ray, Conceptualization, Data curation, Formal analysis, Supervision, Funding acquisition, Validation, Investigation, Visualization, Methodology, Writing – original draft, Project administration; Aminata Toure, Conceptualization, Data curation, Formal analysis, Funding acquisition, Validation, Investigation, Methodology, Writing – original draft, Project administration, Writing – review and editing; Mélanie Bonhivers, Conceptualization, Data curation, Formal analysis, Funding acquisition, Validation, Investigation, Methodology, Writing – original draft, Project administration; Charles Coutton, Conceptualization, Resources, Data curation, Formal analysis, Supervision, Funding acquisition, Validation, Investigation, Visualization, Methodology, Writing – original draft, Project administration, Writing – review and editing

**Author ORCIDs**
Guillaume Martinez http://orcid.org/0000-0002-7572-9096
Derrick Roy Robinson http://orcid.org/0000-0001-5572-4127
Jessica Escoffier http://orcid.org/0000-0001-8166-5845
Christophe Arnoult http://orcid.org/0000-0002-3753-5901
Aminata Toure https://orcid.org/0000-0001-5629-849X
Mélanie Bonhivers http://orcid.org/0000-0001-9179-8473
Charles Coutton http://orcid.org/0000-0002-8873-8098

**Ethics**
Written informed consent was obtained from all individuals participating in the study, and institutional approval was given by the local medical ethics committee (CHU Grenoble Alpes institutional review board #6705). Samples were stored in the Fertithéque collection registered with the French Ministry of Health (DC-2015-2580) and the French Data Protection Authority (DR-2016-392).

Reviewer #1 (Public Review): https://doi.org/10.7554/eLife.87698.3.sa1
Reviewer #2 (Public Review): https://doi.org/10.7554/eLife.87698.3.sa2
Reviewer #3 (Public Review): https://doi.org/10.7554/eLife.87698.3.sa3
Author Response https://doi.org/10.7554/eLife.87698.3.sa4

# Additional files

## Supplementary files
• Supplementary file 1. Proteins co-immunoprecipitated with TbTAX-1 and TbTTC29 identified by mass spectrometry analyses.
• Supplementary file 2. Proteins identification by tandem mass spectrometry (MS/MS) analysis of

proteins co-immunoprecipitated with TTC29 from wild-type and Ttc29$^{-/-}$ L5 and L7 mouse testes.

- Supplementary file 3. Primer sequences for Sanger sequencing verification of *ZMYND12* variants.
- Supplementary file 4. *ZMYND12* MLPA probes used in this work.
- Supplementary file 5. Primers used for quantitative real-time RT-PCR (RT-qPCR) detection of *ZMYND12* in human tissue extracts.
- Supplementary file 6. Primary antibodies used in immunofluorescence experiments with human samples.
- Supplementary file 7. Primary antibodies used in immunofluorescence experiments with *Trypanosoma* cells.
- MDAR checklist

## Data availability

All data generated or analyzed during this study are included in the manuscript and supporting file; Source Data files have been provided for Figures 3–5 and Figure 4-figure Supplement 1. All of the ZMYND12 variants identified here have been deposited in ClinVar under reference SUB12540308. Mass spectrometry proteomics data have been deposited to the ProteomeXchange Consortium via the PRIDE partner repository, under dataset identifier PXD039470 and PXD039854.

The following datasets were generated:

| Author(s) | Year | Dataset title | Dataset URL | Database and Identifier |
|---|---|---|---|---|
| Dupuy JW, Coutton C | 2023 | Enrichment and mass spectrometry analysis of TbTAX-1-interacting or TbTTC29-interacting proteins by co-immunoprecipitation | https://proteomecentral.proteomexchange.org/cgi/GetDataset?ID=PXD039470 | ProteomeXchange, PXD039470 |
| Touré A | 2023 | Identification of TTC29 protein partners in mouse testis | https://proteomecentral.proteomexchange.org/cgi/GetDataset?ID=PXD039854 | ProteomeXchange, PXD039854 |
| Ray PF, Thierry Mieg N, Kherraf Z-E, Coutton C | 2023 | ZMYND12 mutations in MMAF patients | https://www.ncbi.nlm.nih.gov/clinvar/?term=SUB12540308 | ClinVar, SUB12540308 |

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
