## [Editor Report · eLife assessment]

This **important** study reports the physiological role of ZMYND21 in the regulation of sperm flagellar development and male fertility. The data supporting the conclusion are **solid**, although the inclusion of more patients and ultrastructural studies would have further strengthened the study. This work will be of interest to clinicians and researchers who work on either sperm biology or ciliopathy due to cilial defects.

---

## [Referee Report · Reviewer #1 (Public Review)]

The goal of the authors is to use whole-exome sequencing to identify genomic factors contributing to asthenoteratozoospermia and male infertility. Using whole-exome sequencing, they discovered homozygous ZMYND12 variants in four unrelated patients. They examined the localization of key sperm tail components in sperm from the patients. To validate the findings, they knocked down the ortholog in *Trypanosoma brucei*. They further dissected the complex using co-immunoprecipitation and comparative proteomics with samples from Trypanosoma and Ttc29 KO mice. They concluded that ZMYND12 is a new asthenoteratozoospermia-associated gene, bi-allelic variants of which cause severe flagellum malformations and primary male infertility.

The major strengths are that the authors used the cutting-edge technique, whole-exome sequencing, to identify genes associated with male infertility, and used a new model organism, *Trypanosoma brucei* to validate the findings, together with other high-throughput tools, including comparative proteomics to dissect the protein complex essential for normal sperm formation/function. The major weakness is that limited samples could be collected from the patients for further characterization by other approaches, including western blotting and TEM.

In general, the authors achieved their goal, and the conclusion is supported by their results. The findings not only provide another genetic marker for the diagnosis of asthenoteratozoospermia but also enrich the knowledge of cilia/flagella.

---

## [Referee Report · Reviewer #2 (Public Review)]

The manuscript by Dacheux et al. reported homozygous deleterious variants of ZMYND12 in four unrelated men with asthenoteratozoospermia. Based on the immunofluorescence assays in human sperm cells, it was shown that ZMYND12 deficiency altered the localization of DNAH1, DNALI1, WDR66 and TTC29 (four of the known key proteins involved in sperm flagellar formation). *Trypanosoma brucei* and mouse models were further employed for mechanistic studies, which revealed that ZMYND12 is part of the same axonemal complex as TTC29 and DNAH1. Their findings are solid, and this manuscript will be very informative for clinicians and basic researchers in the field of human infertility.

---

## [Referee Report · Reviewer #3 (Public Review)]

In this study, the authors identified homozygous ZMYND12 variants in four unrelated patients. In sperm cells from these individuals, immunofluorescence revealed altered localization of DNAH1, DNALI1, WDR66, and TTC29. Axonemal localization of ZMYND12 ortholog TbTAX-1 was confirmed using the *Trypanosoma brucei* model. RNAi knock-down of TbTAX-1 dramatically affected flagellar motility, with a phenotype similar to ZMYND12-variant-bearing human sperm. Co-immunoprecipitation and ultrastructure expansion microscopy in *T. brucei* revealed TbTAX-1 to form a complex with TTC29. Comparative proteomics with samples from Trypanosoma and Ttc29 KO mice identified a third member of this complex: DNAH1. The data presented revealed that ZMYND12 is part of the same axonemal complex as TTC29 and DNAH1, which is critical for flagellum function and assembly in humans, and Trypanosoma. The manuscript is informative for the clinical and basic researchers in the field of spermatogenesis and male infertility.

---

## [Author Response]

The following is the authors’ response to the original reviews.

**eLife assessment:**
This important study used a battery of cutting-edge technologies including whole exosome sequencing, knockout/knockdown animal models and comparative proteomics to define the physiological roles of ZMYND21 in the regulation of sperm flagellar development and male fertility. The data supporting the conclusion are solid, although inclusion of more patients and ultrastructural studies would have further strengthened the study. This work will be of interest to clinicians and researchers who work on male fertility, but also those working on organs/systems containing motile cilia (e.g., trachea, oviduct, ventricular ependymal cells).

We thank the eLife editorial board for these very positive comments.

The MMAF sperm phenotype is rare and, as for all rare diseases, the number of affected patients remains low. Moreover, the most prevalent genes have already been identified. In such case, the identification of four unrelated patients with pathogenic mutations in the same new gene is thus significant, especially as compared to most studies on the same phenotype. We agree that ultrastructural studies could provide valuable information. However, the amount of sperm cells available did not allow us to consider such experiments at this time. The production and study of the Trypanosoma enabled us to overcome these limitations.

**Reviewer #1 (Public Review):**
The goal of the authors is to use whole-exome sequencing to identify genomic factors contributing to asthenoteratozoospermia and male infertility. Using whole-exome sequencing, they discovered homozygous ZMYND12 variants in four unrelated patients. They examined the localization of key sperm tail components in sperm from the patients. To validate the findings, they knocked down the ortholog in *Trypanosoma brucei*. They further dissected the complex using coimmunoprecipitation and comparative proteomics with samples from Trypanosoma and Ttc29 KO mice. They concluded that ZMYND12 is a new asthenoteratozoospermia-associated gene, biallelic variants of which cause severe flagellum malformations and primary male infertility.The major strengths are that the authors used the cutting-edge technique, whole-exome sequencing, to identify genes associated with male infertility, and used a new model organism, *Trypanosoma brucei* to validate the findings; together with other high-throughput tools, including comparative proteomics to dissect the protein complex essential for normal sperm formation/function. The major weakness is that limited samples could be collected from the patients for further characterization by other approaches, including western blotting and TEM. In general, the authors achieved their goal and the conclusion is supported by their results. The findings not only provide another genetic marker for the diagnosis of asthenoteratozoospermia but also enrich the knowledge in cilia/flagella.

We thank the reviewer for these positive comments that are helpful for improving our paper. Concerning the remark about the low amount of sperm cells available, most patients allowed us to use excess sperm samples not used for ART treatment but are generally reluctant to perform a new sperm collection. Therefore, we often have to prioritize the most relevant and suitable experiments with the amount of sperm cells available.

**Reviewer #2 (Public Review):**
The manuscript "Novel axonemal protein ZMYND12 interacts with TTC29 and DNAH1, and is required for male fertility and flagellum function" by Dacheux et al. interestingly reported homozygous deleterious variants of ZMYND12 in four unrelated men with asthenoteratozoospermia. Based on the immunofluorescence assays in human sperm cells, it was shown that ZMYND12 deficiency altered the localization of DNAH1, DNALI1, WDR66 and TTC29 (four of the known key proteins involved in sperm flagellar formation). *Trypanosoma brucei* and mouse models were further employed for mechanistic studies, which revealed that ZMYND12 is part of the same axonemal complex as TTC29 and DNAH1. Their findings are solid, and this manuscript will be very informative for clinicians and basic researchers in the field of human infertility.

We thank the reviewer for these positive comments that are helpful for improving our paper.

**Reviewer #3 (Public Review):**
In this study, the authors identified homozygous ZMYND12 variants in four unrelated patients. In sperm cells from these individuals, immunofluorescence revealed altered localization of DNAH1, DNALI1, WDR66, and TTC29. Axonemal localization of ZMYND12 ortholog TbTAX-1 was confirmed using the *Trypanosoma brucei* model. RNAi knock-down of TbTAX-1 dramatically affected flagellar motility, with a phenotype similar to ZMYND12-variant-bearing human sperm. Co-immunoprecipitation and ultrastructure expansion microscopy in *T. brucei* revealed TbTAX-1 to form a complex with TTC29. Comparative proteomics with samples from Trypanosoma and Ttc29 KO mice identified a third member of this complex: DNAH1. The data presented revealed that ZMYND12 is part of the same axonemal complex as TTC29 and DNAH1, which is critical for flagellum function and assembly in humans, and Trypanosoma. The manuscript is informative for the clinical and basic research in the field of spermatogenesis and male infertility.

We thank the reviewer for these positive comments that are helpful for improving our paper.

**Reviewer #1 (Recommendations For The Authors):**
The manuscript was very well written, and very easy to follow. Most data were presented in high quality. I only have a few minor issues with some figures.1. The signals in some IF images Fig 1E, Fig. 2B are too weak;

The figures were improved and modified accordingly.

1. In some IF images, strong dot-like signals are observed (Fig. 1B, Fig. 2D, Fig. 2F). Are they specific signals or non-specific? Please specify. If they are non-specific, please replace these images.

These figures were improved and modified accordingly. Indeed, the dot-like signals were non-specific.

**Reviewer #2 (Recommendations For The Authors):**
Here further revisions are suggested.1. Description of ZMYND12 genotypes of the patients and the sperm cell samples:

It was done as suggested

In the Abstract ("with a phenotype similar to ZMYND12-variant-bearing human sperm"), it is suggested to use "with a phenotype similar to the sperm from men bearing homozygous ZMYND12 variants", since the sperm phenotypes are dependent on the biallelic genotypes of human individuals (not the monoallelic genotype of the sperm cells). Please check the whole manuscript and revise the similar points.

It was done as suggested

1. The database accession number for ZMYND12:

It was done as suggested

1. For the exonic deletion variant, is it possible to predict the coding consequence of ZMYND12 protein?

No serious and reliable in silico prediction could be perform due to the absence of the exact breakpoints of the exon deletion. mRNA (or WB) studies could precise this point, however no additional sperm samples from this patient was available.

1. Please italicize the gene symbols. For example, TTC29 on Page 8 and Figure S4, Ttc29-/- KO on Page 13.

It was done as suggested

1. In Figure 2, there are too many panels that cannot be merged into one page. Some of the data can be shown as supplemental data.

We modified the figure 2 as suggested. The new figure 2 now includes only four panels (A, B, C and D) and we added a new figure S4 with the two remaining panels. We modified the text, figure legends and numeration accordingly.

1. Some of the references are duplicated. Please delete one of them.

Sorry for the duplicates. It was corrected

1. The information on some references is incomplete (missing volume and/or page numbers). For example, Touré et al and Wang et al. (2010).

It was corrected

**Reviewer #3 (Recommendations For The Authors):**
However, I have several points as the following:1. The sperm concentrations of ZMYND12_3 in patient 3 and patient 4 are significantly different from the other two patients. Do you think it is just due to phenotype heterogeneity?

We have no formal explanations about these observations but we think that such difference in sperm concentration are more likely due to patient heterogeneity.

1. There is no record for detailed semen parameters of ZMYND12_ 4, and readers cannot see that the proportion of short flagella in Table 1 is 70%. Please provide complete semen routine information for this case.

Unfortunately, no additional information about the semen parameters of this patient are available at this time.

1. In this study, no immunostaining for DNAH1, DNALI1, or WDR66 was detected in sperm from individual ZMYND12_3, and subsequent validation found that TTC29 interacted with ZMYND12 in *Trypanosoma brucei*. DNAH1 and DNALI1 both interact with TTC29 in mice. The author concluded that ZMYND12 is part of the same axonemal complex as TTC29 and DNAH1 and plays a critical role in flagellum function and assembly. If it is possible, the author can add an experiment on the interaction between ZMYND12 and DNAH1 to make this theory more complete.

Our study focuses on characterizing protein-protein interactions using IPs (Immunoprecipitations). We were able to demonstrate that the protein ZYMIND12, along with TTC29, DNAH1, and DNALI1, belongs to the same complex, IAD-4. However, this technique does not allow us to draw conclusions about direct interactions for any of the identified proteins.

Our Co-IP results in *T. brucei* indicate that the orthologue of DNAH1 (Tb927.11.8160 orthologs) and TTC29 co-immunoprecipitate with TAX-1 (ZYMIND12), thereby complementing the study conducted in Chlamydomonas by Yamamoto et al., 2008. As suggested by reviewer 3, direct interactions between each protein could provide valuable insights into the organization of the intracomplex protein interactome. This aspect will be addressed in a separate study, as it requires the use of direct interaction techniques such as Y2H (Yeast Two-Hybrid) or DuoLink.

1. Please check the reference section. Some references have duplication, and the content of the literature also needs to be standardized. For example,Broadhead R., Dawe HR, Farr H, Griffiths S, Hart SR, Portman N, Shaw MK, Ginger ML, Gaskell SJ, McKean PG, Gull K. 2006. Flagellar motility is required for the viability of the bloodstream trypanosome. Nature 440:224-7.Broadhead Richard, Dawe HR, Farr H, Griffiths S, Hart SR, Portman N, Shaw MK, Ginger ML, Gaskell SJ, McKean PG, Gull K. 2006. Flagellar motility is required for the viability of the bloodstream trypanosome. Nature 440:224-227. doi:10.1038/nature04541Ersfeld K, Gull K. 2001a. Targeting of cytoskeletal proteins to the flagellum of *Trypanosoma brucei*. J Cell Sci 114:141-148.Ersfeld K, Gull K. 2001b. Targeting of cytoskeletal proteins to the flagellum of *Trypanosoma brucei*. J Cell Sci 114:141-148. doi:10.1242/jcs.114.1.141

Sorry for the duplicates, it was corrected.